# LOCAL COEFFICIENT OPTIMIZATION IN FEDERATED LEARNING

## ABSTRACT

Federated learning emerges as a promising approach to build a large-scale co-operative learning system among multiple clients without sharing their raw data. However, given a specific global objective, finding the optimal sampling weights for each client remains largely unexplored. This is particularly challenging when clients' data distributions are non-i.i.d. and clients partially participant.

In this paper, we model the above task as a bi-level optimization problem which takes the correlations among different clients into account. We present a double-loop primal-dual-based algorithm to solve the bi-level optimization problem. We further provide rigorous convergence analysis for our algorithm under mild assumptions. Finally, we perform extensive empirical studies under both toy examples and learning models from real datasets to verify the effectiveness of the proposed method.

## 1 INTRODUCTION

Federated learning has achieved high success in the large-scale cooperative learning system without sharing raw data. However, due to the large number of devices involved in the learning system, it is hard to check data quality (e.g., noise level) for individual devices. Further, it will degrade the model's ability when it is trained with bad-quality data. To eliminate the influence of the 'bad' devices, it is natural to reduce the weight of those devices. In most popular federated training algorithms (e.g., FedAvg (Li et al., 2019)), all devices are weighted the same or with respect to the number of data points it holds. Borrowing the formulation of federated algorithms, we introduce a new variable $x$ to control the weight of each device which is the coefficient of each local objective. We introduce a validation set in the server to validate whether coefficients improve the model. We formulate the whole problem as a bi-level optimization state in the following:

$$
\begin{aligned}
\min_{x} \quad & f_0(w^*(x)) \\
\text{s.t.} \quad & w^*(x) \in \arg\min_{w} \sum_{i=1}^{N} x^{(i)} f_i(w) \\
& x \in \mathcal{X} = \{x | x \geq 0, \|x\|_1 = 1\},
\end{aligned}
\tag{1}
$$

To solve problem (1), Kolstad & Lasdon (1990) propose an algorithm that calculates the gradient of $x$ directly, i.e.,

$$
\frac{\partial f_0(w^*(x))}{\partial x^{(i)}} = -\nabla_w f_0(w^*(x))^\top \left( \sum_{i=1}^{N} \nabla_w^2 f_i(w^*(x)) \right)^{-1} \nabla_w f_i(w^*(x)).
$$

But, due to the large parameter dimension of $w$, it is impossible to take the inverse of the Hessian or solve the linear system related to the Hessian. Meanwhile, due to a large amount of data in the local device, it is hard to directly estimate the gradient or the Hessian of the local function $f_i$. Only stochastic gradient and stochastic hessian can be accessed. Thus, Ghadimi & Wang (2018) propose the BSA algorithm where the inverse of Hessian is approximated by a series of the power of Hessian (using $\sum_{k=0}^{K}(I - \eta H)^k$ to approximate $\frac{1}{\eta} H^{-1}$ with certain $\eta$). Khanduri et al. (2021) propose SUSTAIN algorithm for solving stochastic bi-level optimization problems with smaller

sample complexity. Similar to Ghadimi & Wang (2018), they need an extra loop of the Hessian-vector product to approximate the product of the Hessian inverse with some vector.

However, it is known that for constraint optimization, with the descent direction, the algorithm will not converge to the optimal point or even to the first-order stationary point, where the inner product between the descent direction and gradient is larger than 0 (Bertsekas, 2009). Therefore, getting an accurate approximation of the hessian inverse is essential. With the series of the power, it has to start with $k = 0$ and apply several iterations to get an accurate approximation, increasing the computation and communication in federated learning. Fortunately, by noticing the KKT condition, information of the hessian inverse can be embedded into dual variables. Based on the smoothness of the objective, we can give a good initialization of dual variables rather than start with the same initialization in each iteration (like $I$ in the series approximation). Thus, we propose a primal-dual-based algorithm to solve problem (1).

Further, to solve a constrained optimization with non-linear equality constraints, adding the norm square of the equality constraint as an augmented term may not give the convexity to the augmented Lagrange function. As a result, it is hard for the min-max optimization algorithm to find the stationary point. Instead, with the assumption in Ghadimi & Wang (2018), the function $f_i$'s are assumed to be strongly convex, adding function $f_i$'s as the augmented term can help introduce convexity and it will not change the stationary point of the min-max problem. Based on this new augmented Lagrange function, we prove that with stochastic gradient descent and ascent, $w$ and $\lambda$ can converge to the KKT point. Meanwhile, by the implicit function theorem, when $w$ and $\lambda$ are closed to the stationary point of min-max, the bias of estimating the gradient of $x$ can be reduced to 0. Thus, with the primal-dual algorithm on $w$ and $\lambda$ and stochastic projected gradient descent on $x$, we show the convergence of our algorithm.

Finally, we compare our algorithm with other algorithms on a toy example and real datasets (MNIST and F-MNIST with Network LeNet-5). The experimental results show that the proposed algorithm can perform well in strongly convex cases and even in some non-convex cases (Neural Networks).

We summarize our contributions as follows:

- In Federated Learning, we formulate the local coefficient learning problem as a bi-level optimization problem, which gives a way to identify the dataset quality in each local client for some specific task (where a small validation set is given).
- In bi-level optimization, we introduce a primal-dual framework and show the convergence of the whole algorithm in the constrained and stochastic setting.
- For some specific optimization problems with non-linear constraints, we give a new augmented term. With the new augmented term, the primal variable and dual variable can converge to the KKT point of the original problems.

## 2 RELATED WORK

### 2.1 PERSONALIZED FEDERATED LEARNING

The most related work in federated learning tasks will be personalized federated learning. A well-trained local personalized personalized model is needed for each local device in personalized federated learning. Jiang et al. (2019); Deng et al. (2020) propose a method that they train a global model and then fine-tune the trained global model to get the local model. T Dinh et al. (2020); Fallah et al. (2020) change the local objective function to make each local has the ability to be different and handle individual local tasks. Li et al. (2021) introduces a two-level optimization problem for seeking the best local model from great global models. All of these works do not involve a validation set as a reference, but they use a few gradient steps or simple modifications and hope the local model can both fit the local training data and use information from the global model (other local devices). Different from these works, we explicitly formulate a bi-level optimization problem. By adding a validation set, it can be more clearly identified the correlation of information from the other devices and from its own.

### 2.2 STOCHASTIC BI-LEVEL OPTIMIZATION

Bi-level optimization problem has been studied for a long time. One of the simplest cases in bi-level optimization is the singleton case, where the lower-level optimization has a unique global optimal

point. Without calculating the inversion of the Hessian matrix of the lower level optimization problem, there are two major algorithms. Franceschi et al. (2017) approximates $\frac{\partial w^*(x)}{\partial x}$ by $\frac{\partial w_T}{\partial x}$ where $w_T$ is the iterate after T steps gradient descent for the lower optimization problem. Using this method, in each iteration, we need to communicate $N$ (number of local devices) vectors among the server and local devices which is not communication efficient. The other method Ghadimi & Wang (2018) is to approximate $(\nabla_w^2 g(w))^{-1}$ by $\sum_{i=0}^{K}(I - \eta\nabla^2 g(w))^i$, where $g(w)$ is the objective function of lower-level optimization problem. Although Khanduri et al. (2021) point out that to approximate gradient for upper optimization function, we can get rid of taking the optimal point for lower optimization in each upper-level update optimization, which seems to get rid of double-loop approximation, it still needs a loop for approximating Hessian inverse with series. Guo & Yang (2021) uses SVRG to reduce the noise level of estimating stochastic gradient and Hessian to get better performance. Besides, all of the above works assume smoothness of the local Hessian, but none of them will apply the property directly into the algorithm. Different from the above works, we introduce a primal-dual framework into bi-level optimization, where the dual variable can record the information of Hessian. Also, Shi et al. (2005); Hansen et al. (1992) introduce the primal-dual framework, but they stay in quadratic regime or mix integer programming, which is non-trivial to extend the results to federated learning settings.

## 3 ALGORITHM DESIGN

Assume that each function of $f_i$ is a strongly convex function. Then, the optimal solution to the lower optimization problem becomes only a single point. Thus, with the implicit function theorem, we can calculate the gradient of $f_0(w^*(x))$ with respect to $x$ as follows.

**Proposition 1.** *Suppose $f_i$'s are strongly convex functions. Then for each $x \in \mathcal{X}$, it holds that*
$\frac{\partial f_0(w^*(x))}{\partial x^{(i)}} = -\nabla_w f_0(w^*(x))^\top \left(\sum_{j=1}^{N} x^{(j)} \nabla_w^2 f_j(w^*(x))\right)^{-1} \nabla_w f_i(w^*(x)).$

With the proposition 1, one can calculate the gradient of $x$, when $w^*(x)$ and the inverse of Hessian are given. However, for large scale problems, none of these can be easily obtained. Fortunately, by noticing the convexity of each function $f_i$, we can replace the first constraint $w^*(x) \in \arg\min_w \sum_{i=1}^{N} x^{(i)} f_i(w)$ with $\nabla \sum_{i=1}^{N} x^{(i)} f_i(w) = 0$. For given $x$, we can formulate the following constrained optimization problem:

$$\min_w \ f_0(w)$$
$$s.t. \ \sum_{i=1}^{N} x^{(i)} \nabla_w f_i(w) = 0, \tag{2}$$

By introducing the dual variable $\lambda$, we can easily get the Lagrange function. To solve the Lagrange function efficiently, we propose the following augmented Lagrange function.

$$L_x(w, \lambda) = f_0(w) + \lambda^\top \sum_{i=1}^{N} x^{(i)} \nabla_w f_i(w) + \Gamma \sum_{i=1}^{N} x^{(i)} f_i(w). \tag{3}$$

Different from the standard augmented terms, where the norm square of equality constraints is added to achieve strong convexity of the primal problem, we add the summation of $f_i$'s with coefficient $x^{(i)}$'s. If we use the norm square of the gradient constraint for general strongly convex functions, it will not be strongly convex. Thus, we can not directly adopt the gradient descent ascent algorithm. With the definition, we can obtain the following two propositions directly.

**Proposition 2.** *Suppose $f_i$'s are strongly convex functions for $i = 1, 2, \cdots, N$, $x^{(i)} \geq 0$ for all $i$ and $\|x\|_1 = 1$. Then, Problem (2) satisfies Linear Independence Constraint Qualification and its KKT conditions can be written as follows:*

$$\nabla_w f_0(w) + \sum_{i=1}^{N} x^{(i)} \nabla^2 f_i(w)\lambda = 0$$
$$\sum_{i=1}^{N} x^{(i)} \nabla f_i(w) = 0.$$

**Proposition 3.** *Suppose $f_i$'s are strongly convex functions for $i = 1, 2, \cdots, N$, $x^{(i)} \geq 0$ for all $i$ and $\|x\|_1 = 1$. Then, the stationary point of $\min_w \max_\lambda L_x(w, \lambda)$ is unique and satisfies the KKT conditions of problem* (2).

Let $(\hat{w}^*(x), \lambda^*(x))$ be the stationary point of $\min_w \max_\lambda L_x(w, \lambda)$. From proposition 2, it holds that $\hat{w}^*(x) = w^*(x)$ and

$$\frac{\partial f_0(w^*(x))}{\partial x^{(i)}} = \lambda^*(x)^\top \nabla_w f_i(w^*(x)). \tag{4}$$

Thus, with the KKT poiont $w^*(x)$ and $\lambda^*(x)$, we can estimate the gradient of $x$ without estimating the inverse of Hessian. However, as $\lambda^\top \sum_{i=1}^N x^{(i)} \nabla_w f_i(w)$ can be highly non-convex function, which can be harmful to the optimization process. We add an additional constraint on the norm of $\lambda$ and define the constraint set $\Lambda$. Thus, the problem (2) becomes

$$\min_w \max_{\lambda \in \Lambda} L_x(w, \lambda) = f_0(w) + \lambda^\top \sum_{i=1}^N x^{(i)} \nabla_w f_i(w) + \Gamma \sum_{i=1}^N x^{(i)} f_i(w). \tag{5}$$

We propose a double loop algorithm for solving problem (1). We show the algorithm in the Algorithm 1 and 2. In inner loop, we solve the augmented Lagrange for $K$ steps. In each step, local client will receive the iterates $w_{t,k}$ and $\lambda_{t,k}$. After that, each local client will calculate $\tilde{\nabla} f_i(w_{t,k})$ and $\hat{\nabla} f_i(w_{t,k})$ based on the back propagation through two independent batches. The term $\tilde{\nabla}^2 f_i(w_{t,k}) \lambda_{t,k}$ is calculated with auto-differentiable framework (i.e. Pytorch, TensorFlow) or with the closed-form multiplication. Then the local device sends gradient estimation $\tilde{\nabla}_w f_i(w_{t,k})$ and the estimated product of Hessian and $\lambda$ ($\tilde{\nabla}^2 f_i(w_{t,k}) \lambda_{t,k}$) to the server.

For the server, in each step, the server will first send the primal variable $(w_{t,k})$ and dual variable $(\lambda t, k)$ to all local clients. Then, the server will receive the estimated gradients and estimated product from some local clients. Because not all devices will stay online in each step, we define a set $Active_{t,k}$ which records the clients that participant the optimization in $(t, k)$ step. With the vectors collected from local clients, the server will calculate the gradient estimator of $w_{t,k}$ and $\lambda_{t,k}$ with respect to function $L_{x_t}(w_{t,k}, \lambda_{t,k})$. And then, $w_{t,k}$ will be updated by a gradient descent step and $\lambda_{t,k}$ will be updated by a gradient ascent step. Different from local devices, after K inner loop update steps, based on the $\lambda_{t,K}$ and gradient estimated in each local client, the server will calculate the gradient of $x$ based on equation 4 and perform a projected gradient descent step on $x$. In addition, if the $i_{\text{th}}$ agent is not in $Active_{t,K}$, we set the gradient of $x^{(i)}$ to be zero.

---

**Algorithm 1** The bi-level primal dual algorithm on local device $i$

---

1: **for** $t = 1, 2, \cdots, T$ **do**
2:     **for** $k = 1, 2, \cdots, K$ **do**
3:         Receive $w_{t,k}, \lambda_{t,k}$ from the server;
4:         Sample a mini-batch and calculate $\tilde{\nabla} f_i(w_{t,k})$;
5:         Sample a mini-batch and calculate $\hat{\nabla} f_i(w_{t,k})$;
6:         Calculate $\tilde{\nabla}_w^2 f_i(w_{t,k}) \lambda_{t,k}$ with back propagation on scalar $\hat{\nabla}_w f(w_{t,k}) \lambda_{t,k}$;
7:         Send $\tilde{\nabla}^2 f_i(w_{t,k}) \lambda_t$ and $\tilde{\nabla} f_i(w_{t,k})$ to the server;
8:     **end for**
9: **end for**

---

**Remark 1.** *$g_{x^{(i)}}$ can be calculated in the i-th device and sent to the server, which can reduce the computation in the server and will increase one-round communication with one real number between the server and devices. The rest of the analysis will remain to be the same.*

## 4 THEORETICAL ANALYSIS

In this section, we analyze the convergence property of the proposed algorithm. First, we state some assumptions used in the analysis.

**(A1)** $f_0, f_1, \cdots, f_N$ are lower bounded by $\underline{f}$, and $f_0, f_1, \cdots, f_N$ have $L_1$ Lipschitz gradient.

**(A2)** $f_1, \cdots, f_N$ are $\mu$-strongly convex functions.

---

**Algorithm 2** The Bi-level primal dual algorithm on the Server

1: **Input:** Initial $x_1, w_{1,1}, \lambda_{1,1}$, **total iterations:** $K,\ T$ **and step size:** $\eta_w, \eta_\lambda, \eta_x$.
2: **for** $t = 1, 2, \cdots, T$ **do**
3:     **for** $k = 1, 2, \cdots, K$ **do**
4:         Send $w_{t,k}, \lambda_{t,k}$ to each local device;
5:         Receive $\tilde{\nabla} f_i(w_{t,k})$ and $\tilde{\nabla}_w^2 f_i(w_{t,k})\lambda_t$ from $Active_{t,k}$;
6:         $g_w = \tilde{\nabla} f_0(w_{t,k}) + \frac{N}{|Active_{t,k}|} \sum_{i \in Active_{t,k}} x_t^{(i)} \tilde{\nabla}_w^2 f_i(w_{t,k})\lambda_{t,k} + \Gamma \tilde{\nabla} f_i(w_{t,k})$;
7:         $w_{t,k+1} = w_{t,k} - \eta_w g_w$;
8:         $g_\lambda = \frac{N}{|Active_{t,k}|} \left( \sum_{i \in Active_{t,k}} x_t^{(i)} \tilde{\nabla}_w f_i(w_{t,k}) \right)$;
9:         $\lambda_{t,k+1} = \Pi_\Lambda \left( \lambda_{t,k} + \eta_\lambda g_\lambda \right)$;
10:    **end for**
11:     $g_{x^{(i)}} = \frac{N}{|Active_{t,K}|} \lambda_{t,K}^\top \tilde{\nabla}_w f_i(w_{t,K})$ for $i \in Active_{t,K}$;
12:     $g_{x^{(i)}} = 0$ for $i \notin Active_{t,K}$;
13:     $x_{t+1} = P_\mathcal{X}(x_t - \eta_x g_x)$;
14:     $\lambda_{t+1,1} = \lambda_{t,K+1}$;
15:     $w_{t+1,1} = w_{t,K+1}$
16: **end for**
17: **Output:** $x_T, W_{T,K+1}$.

---

(A3) $f_1, \cdots, f_N$ has $L_2$ Lipschitz Hessian.

(A4) $\max_{i \in \{0,1,\cdots,N\}} \max_{x \in \mathcal{X}} \|\nabla f_i(w^*(x))\| \leq D_w$.

(A5) Each local estimation is unbiased with bounded variance $\sigma^2$.

(A6) $Active_{t,k}$ is independent and sampled from the set of nonempty subset of $\{1, 2, \cdots, N\}$, where $P(i \in Active_{t,k}) = p$ for all $i \in \{1, 2, \cdots, N\}$.

**Remark 2.** *(A1),(A2),(A3) are commonly used in the convergence analysis for bi-level optimization problems (Ji et al., 2021; Chen et al., 2021; Khanduri et al., 2021). Unlike Ji et al. (2021); Chen et al. (2021), where they need to assume $f_0, f_1, \cdots, f_N$ to be $L_0$ Lipschitz, we assume the gradient norm are bounded at optimal solution. Because for machine learning models, regularization will be add into objective function, makes the norm of the optimal solution not be large. When $w^*(x)$ can be bounded by some constant. (A4) is reasonable in practice. Moreover, the Lipschitz assumption on function can directly infer (A4) with $D_w = L_0$. (A5) is a common assumption used for stochastic gradient methods (Ghadimi et al., 2016) and (A6) extend the assumption in Karimireddy et al. (2020) by giving the probability that a local devices will be chosen instead of uniformly sampling.*

**Remark 3.** *With (A4), $D_\lambda = \max_{x \in \mathcal{X}} \|\lambda^*(x)\|$ is upper bounded by $D_w/\mu$.*

**Proposition 4.** *When $\Lambda = \{\lambda \mid \|\lambda\| \leq D_\lambda\}$, then the stationary point of problem (5) is the KKT point of problem 2.*

With proposition 3 and 4, the stationary point of problem (5) is unique and we denote the stationary point as $(w^*(x), \lambda^*(x))$. To give the convergence of the whole algorithm, firstly, we give the convergence guarantee for the inner loop.

**Theorem 1.** *For given $x \in \mathcal{X}$, when (A1) to (A6) holds, $\Gamma > \frac{D_\lambda L_2 + L_1}{\mu}$ and $\eta_w, \eta_\lambda = \Theta(1/\sqrt{K})$, when randomly choose $\hat{k} \in \{1, 2, \cdots, K\}$ with equal probability it holds that*

$$\mathbb{E}\left[ \left\| \lambda_{\hat{k}}^\top \nabla_w f_i(w_{\hat{k}}) - \frac{\partial f_0(w^*(x))}{\partial x^{(i)}} \right\|^2 \right] = O(1/\sqrt{K}).$$

Thus, with theorem 1, the gradient of $x$ can be "well" estimated through the inner gradient descent ascent method when the number of inner loop steps is large enough. Then, we can obtain the following convergence result of the outer loop.

**Theorem 2.** *Suppose (A1) to (A6) holds, $\Gamma > \frac{D_\lambda L_2 + L_1}{\mu}$, $\eta_w, \eta_\lambda = \Theta(1/\sqrt{K})$, $\eta_x = \Theta(1/\sqrt{T})$ and randomly choosing $\hat{k} \in \{1, 2, \cdots, K\}$ with equal probability to approximate gradient of x. Define*

$\hat{x} = \arg\min_{y \in \mathcal{X}} (f_0(w^*(y)) + \frac{\rho}{2}\|y - x\|^2)$ *and* $\bar{\nabla}_\rho f_0(w^*(x)) = \rho(x - \hat{x})$ *for large* $\rho$, *it holds that*

$$\frac{1}{T}\sum_{t=1}^{T} \mathbb{E}\|\bar{\nabla}_\rho f_0(w^*(x_t))\|^2 = O(1/\sqrt{T} + 1/\sqrt{K}).$$

**Remark 4.** *To achieve $\epsilon$-stationary point ($\mathbb{E}\|\bar{\nabla}_\rho f_0(w^*(x_t))\|^2 \leq \epsilon$), $O(1/\epsilon^4)$ samples are needed in each local client and in the server. Different from the* previous works on bilevel optimization*(e.g. Ghadimi & Wang (2018), Khanduri et al. (2021) and Franceschi et al. (2017)), we prove the convergence when optimization variable $x$ has a convex constraint.*

## 4.1 PROOF SKETCH OF THEOREM 1

To show the convergence of inner loop, we first construct a potential function for inner loop objective. Define $\Phi_x(w, \lambda) = L_x(w, \lambda) - 2d(\lambda)$, where $d(\lambda) = \min_w L_x(w, \lambda)$ for given x. The intuition of defining this potential function is that $L_x(w, \lambda)$ is not necessarily decreasing in each iteration, as $\lambda$ is performing a gradient ascent step. Meanwhile, gradient $\lambda$ taken is an approximation of gradient of $d(\lambda)$. Thus, by subtracting $d(\lambda)$, we can obtain that $\Phi$ will decrease during iterations. Therefore, the first thing is to show the lower bound of function $\Phi$.

**Lemma 1** (Lower bound of $\Phi$). *Suppose **(A1)-(A4)** hold. It holds that $\Phi_x(w, \lambda)$ is bounded below by $\underline{f}$.*

The proof of this lemma is basically due to the definition of $\Phi_x(w, \lambda)$ and $d(\lambda)$. Then, similar to the proof of gradient descent, we give a lemma that shows the descent of potential function under certain choices of hyperparameters.

**Lemma 2** (Potential function descent, proof can be found in Lemma 11 in Appendix). *Suppose **(A1)-(A6)** hold. In addition, we assume $\Gamma > \frac{D_\lambda L_2 + L_1}{\mu}$, it holds that*

$$\mathbb{E}[\Phi_x(w_{t,k}, \lambda_{t,k}) - \Phi_x(w_{t,k+1}, \lambda_{t,k+1})] \leq -C_1\mathbb{E}\|\nabla_w L_x(w_{t,k}, \lambda_{t,k})\|^2 - C_2\mathbb{E}[\|\lambda_t - \lambda_t^*\|^2] + C_3\sigma^2,$$

*where $\lambda_t^+ = \Pi_\Lambda(\lambda_t + \eta_\lambda \nabla d(\lambda_t))$, $C_1 = \Theta(\eta_w - \eta_w^2 - \eta_\lambda^2 - \eta_\lambda)$, $C_2 = \Theta(\eta_\lambda)$ and $C_3 = O(\eta_w^2 + \eta_\lambda^2)$*

Thus, when choosing sufficient small $\eta_w$ and $\eta_\lambda$, we can achieve positive $C_1$ and $C_2$. Together with the lower bound of the function $\Phi$, the convergence of the inner algorithm can be shown. Because of the uniqueness of the KKT point, by choosing $\eta_w$ and $\eta_\lambda$ in order of $1/\sqrt{K}$, it can be shown that

$$\frac{1}{K}\sum_{k=1}^{K}\mathbb{E}\|w_{t,k} - w^*(x_t)\|^2 = O(1/\sqrt{K}), \quad \frac{1}{K}\sum_{k=1}^{K}\mathbb{E}\|\lambda_{t,k} - w^*(x_t)\|^2 = O(1/\sqrt{K}).$$

Therefore, with the convergence rate of $w_{t,k}$ and $\lambda_{t,k}$ and equation 4, we can easily prove theorem 1.

## 4.2 PROOF SKETCH OF THEOREM 2

To apply stochastic gradient descent analysis on $x$, although we have smoothness for function $f_0, f_1, \cdots, f_N$ on $w$, we need to verify the smoothness of $f_0(w^*(x))$ with respect to $x$.

**Lemma 3** (Convergence of stochastic gradient descent with biased gradient estimation, proof can be found in Lemma 14 in Appendix). *Suppose function $f(x)$ is lower bounded by $\underline{f}$ with $L$-Lipshitz gradient. $g(x)$ is an unbiased gradient estimator of $\nabla f(x)$ satisfying that expected norm of $g(x)$ are bounded by $G$ in domain $\mathcal{X}$ for function $f$. Then with update rule $x_{t+1} = \Pi_\mathcal{X}(x_t - \eta_x(g(x_t) + \xi_t))$, where $\eta_x = \Theta(1/\sqrt{T})$, $\mathcal{X}$ is a convex set and $\mathbb{E}\|\xi_t\|^2 \leq \epsilon^2$. By defining $\hat{x} = \arg\min_{y \in \mathcal{X}}(f(y) + \frac{\rho}{2}\|y - x\|^2)$ and $\bar{\nabla}_\rho f(x) = \rho(x - \hat{x})$, where $\rho = 2L$, then it holds that*

$$\frac{1}{T}\mathbb{E}\sum_{t=1}^{T}\|\bar{\nabla}_\rho f(x_t)\|^2 = O(1/\sqrt{T} + \epsilon^2).$$

As Lemma 3 suggests, when $f_0(w^*(x))$ satisfying $L$-Lipschitz gradient, bounded estimation error and bounded gradient norm, the convergence rate can achieve $O(1/\sqrt{T})$ with a error term related to estimation error. Theorem 1 shows the estimation error can be bounded by $O(1/\sqrt{K})$. Combining this two results we can prove Theorem 2.

## 5    EXPERIMENTAL RESULTS

In this section, we compare our algorithm with other bi-level optimization algorithms (BSA (Ghadimi & Wang, 2018), SUSTAIN (Khanduri et al., 2021) and RFHO (Franceschi et al., 2017)) in two cases: the toy example and two vision tasks. Further, in vision tasks, agnostic federated learning (AFL) is tested (Mohri et al., 2019). When k local steps are used in each algorithm, BSA, RFHO, and our algorithm will perform $2kd$ real number transmission, where $d$ is the dimension of optimization. SUSTAIN will perform $(k+1)d$ real number transmission. In the vision tasks, they perform the same real number of transmissions as $k = 1$.

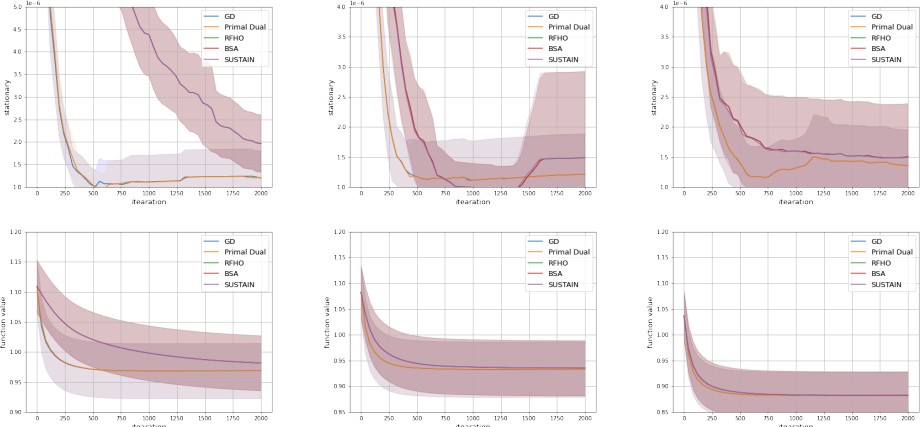

Figure 1: The figure shows the result of the toy example where all clients participate in the optimization process in each iteration, and all gradient and hessian are estimated without noise. The above line shows the stationary of $x$ in each iteration, and the second row shows the function value of $x$ ($f(w^*(x))$). The left column shows the results when the number of local steps is 1; the middle column shows the results of 5 local steps, and the right column gives the results of 10 local steps. The shadow part of the function value corresponds to the 0.1 standard error area, and the shadow part in stationary corresponds to the 0.5 standard error area.

### 5.1    TOY EXAMPLE

In this section, we apply algorithms to solve problem (1) with $f_i$ as follows:

$$f_i(w) = \frac{1}{2}\|A_i w - B_i\|^2 + cos(a_i^\top w - b_i),$$

where $A_i \in \mathbb{R}^{30 \times 20}, B_i \in \mathbb{R}^{30}, a_i \in \mathbb{R}^{20}$ and $b_i \in \mathbb{R}$ are all generated from Gaussian distribution. The variance of each component in $A_i$ and $a_i$ is $1/\sqrt{20}$ and the variance of each component in $B_i$ is $1/\sqrt{30}$ and variance of $b_i$ is 1. When generated function $f_i$ is not 0.1-strongly convex, we randomly generate a new one until we get strongly convex $f_i$ whose modular is not less than 0.1. Three local steps ($K$=1,5,10) are tested. Here, the local steps are used for $w$ update for algorithm BSA, RFHO, and our algorithm, and the local steps are used for Hessian estimation for algorithm BSA and SUSTAIN. Because for this toy example, we can easily compute the Hessian matrix and its inverse, we test the algorithm using the inverse of estimated Hessian to compute the gradient of $x$ named GD. We test two settings of the toy example. One is the deterministic setting, where no estimation noise or client disconnection will occur. In the other setting, we add white Gaussian noise with a noise level of 0.5 in each estimation (including gradient estimation and Hessian estimation). Also, each client has a 0.5 probability of connecting with the server.

To evaluate the performance of different algorithms, we calculate the function value of $f_0(w^*(x))$ and the stationary of x, i.e. $x - \Pi_X(x - 0.001\nabla_x f_0(w^*(x)))$, where $w^*(x)$ is approximate by 200 gradient steps. We take $N = 15$ and run 20 times and get the results of different algorithms. The results of deterministic setting are shown in Figure 1, and results of noise setting are shown in Figure 2.

As it is shown in Figure 1, with local steps getting larger and larger, the performance of BSA, RFHO, and SUSTAIN is getting close to GD, while the performance of the primal-dual method is similar to

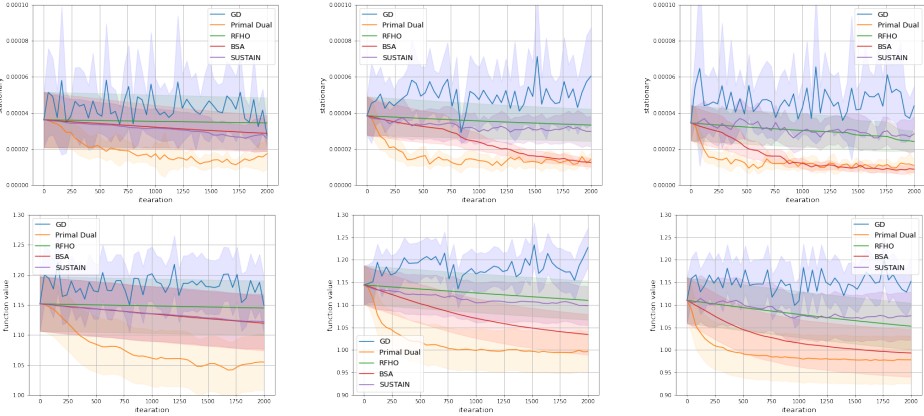

Figure 2: The figure shows the result of the toy example where the active rate is 0.5 in each iteration, and all gradient and hessian are estimated with white-Gaussian noise with a noise level of 0.5. The above line shows the stationary of $x$ in each iteration, and the second row shows the function value of $x$ ($f(w^*(x))$). The left column shows the results when the number of local steps is 1; the middle column shows the results of 5 local steps, and the right column gives the results of 10 local steps. The shadow part of the function value corresponds to the 0.1 standard error area, and the shadow part in stationary corresponds to the 0.5 standard error area.

GD whatever local step used in the algorithm even with only one single step. When noise is added in the Hessian, directly inverse may cause the biased estimation. Thus, the performance of GD gets much worse than it in the deterministic setting shown in Figure 2. Also, in Figure 2, our algorithm can perform better than other algorithms when the local step is small. When local steps increase to 10, BSA and our algorithm have competitive results.

## 5.2 VISION TASKS

Table 1: Test Accuracy and $x$ output of Training LeNet 5 on MNIST. "AP" represents Active Probability, and Accuracy stands for Test Accuracy.

| AP | | RFHO | BSA | SUSTAIN | Ours |
|---|---|---|---|---|---|
| 1 | Accuracy | $98.34\% \pm 0.18\%$ | $98.15\% \pm 0.23\%$ | $99.02\% \pm 0.15\%$ | $98.43\% \pm 0.17\%$ |
| | $x^{(1)}$ | $0.488 \pm 0.104$ | $0.425 \pm 0.081$ | $0.411 \pm 0.069$ | $0.455 \pm 0.016$ |
| | $x^{(2)}$ | $0.311 \pm 0.104$ | $0.245 \pm 0.133$ | $0.305 \pm 0.045$ | $0.334 \pm 0.020$ |
| | $x^{(3)}$ | $0.197 \pm 0.031$ | $0.294 \pm 0.176$ | $0.282 \pm 0.029$ | $0.212 \pm 0.026$ |
| | $x^{(4),\cdots,(10)}$ | $\sim 6e-4$ | $\sim 6e-3$ | $\sim 3e-4$ | $\sim 2e-4$ |
| 0.9 | Accuracy | $98.07\% \pm 0.4\%$ | $98.09\% \pm 0.21\%$ | $98.85\% \pm 0.29\%$ | $98.43\% \pm 0.19\%$ |
| | $x^{(1)}$ | $0.407 \pm 0.040$ | $0.395 \pm 0.136$ | $0.386 \pm 0.058$ | $0.449 \pm 0.046$ |
| | $x^{(2)}$ | $0.281 \pm 0.065$ | $0.314 \pm 0.045$ | $0.345 \pm 0.028$ | $0.333 \pm 0.050$ |
| | $x^{(3)}$ | $0.291 \pm 0.018$ | $0.239 \pm 0.085$ | $0.265 \pm 0.038$ | $0.217 \pm 0.024$ |
| | $x^{(4),\cdots,(10)}$ | $\sim 4e-3$ | $\sim 8e-3$ | $\sim 7e-4$ | $\sim 2e-4$ |
| 0.5 | Accuracy | $97.86\% \pm 0.36\%$ | $95.37\% \pm 4.10\%$ | $97.60\% \pm 0.49\%$ | $98.24\% \pm 0.23\%$ |
| | $x^{(1)}$ | $0.449 \pm 0.090$ | $0.539 \pm 0.076$ | $0.365 \pm 0.015$ | $0.468 \pm 0.052$ |
| | $x^{(2)}$ | $0.276 \pm 0.075$ | $0.217 \pm 0.059$ | $0.329 \pm 0.013$ | $0.372 \pm 0.053$ |
| | $x^{(3)}$ | $0.271 \pm 0.129$ | $0.210 \pm 0.039$ | $0.292 \pm 0.015$ | $0.16 \pm 0.035$ |
| | $x^{(4),\cdots,(10)}$ | $\sim 6e-4$ | $\sim 6e-3$ | $\sim 2e-3$ | $\sim 2e-4$ |

In this section, we apply algorithms to train LeNet5(LeCun et al., 1998) on dataset MNIST(LeCun et al., 1998) and Fashion-MNIST(Xiao et al., 2017). To construct non-iid datasets on different local clients and the global server's validation set, we randomly pick 20 samples per label out of the whole training dataset and form the validation set. Then, the rest of the training data are divided into 3 sets, and each set will be assigned to a local client. The first client contains samples labeled as 0,1,2,3,4, the second client contains samples labeled as 5,6,7, and the third client contains samples labeled as 8,9 for all two datasets. To test the algorithm's ability to choose the proper coefficient of local clients, we add 7 noise nodes containing 5000 samples with random labels. We set the learning rate of $w$ to be a constant learning rate without any decay selected from $\{0.1, 0.01, 0.001\}$ for all training

methods, and the learning rate of $x$ is selected from $\{0.1, 0.01, 0.001, 0.0001\}$. The batch size for all three training cases is set to 64. $\Gamma$ used in the proposed algorithm is set to be 1. For simplicity, we set the local step as 1. We run 2000 iterations for MNIST and 6000 iterations for Fashion-MNIST. Active probability is set in $\{0.5, 0.9, 1\}$. We compare the test accuracy among different methods. As a baseline, we report the test accuracy for training with the validation set only named val, training with the average loss of each client named avg, and training with $x = (0.5, 0.3, 0.2, 0, \cdots, 0)$ named opt. All experiments run on V100 with Pytorch (Paszke et al., 2019). Results are shown in Figure 3, Figure 4 and Table 1.

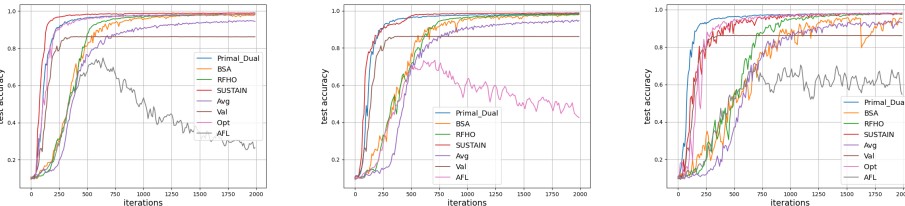

Figure 3: Test accuracy of training LeNet 5 on MNIST dataset. The left curve shows the result when the active probability is 1; the middle curve shows the result when the active probability is 0.9, and the right curve shows the result with the active probability of 0.5.

Figure 3 shows the test accuracy of the MNIST dataset with different active probabilities. Although SUSTAIN works better than the primal-dual algorithm when all local devices participate in the optimization process, when clients' participant rate decreases to 0.5, SUSTAIN works worse than ours method. Primal-dual become slower than SUSTAIN may be because of the initialization of the dual variable. When the dual variable is far from its real value it needs more time to get a good enough point. Other than SUSTAIN, our algorithm can converge faster and more stable to a high accuracy point. Further, we list the output of $x$ and standard error of test accuracy for 5 different experiments for different algorithms in Table 1. According to Table 1, our algorithm can achieve a more stable output with respect to $x$, and the output $x$ is more close to $0.5, 0.3, 0.2$, which is related to the number of labels the first three clients holds.

Figure 4 gives the test accuracy of training LeNet 5 on the Fashion-MNIST Dataset. Similar to the results of the MNIST dataset, when the clients' participant is high (0.9,1), SUSTAIN works slightly better than the primal-dual algorithm. But when more local devices disconnect to the server, the performance of SUSTAIN drops, while the primal dual algorithm remains fast convergence speed and high test accuracy.

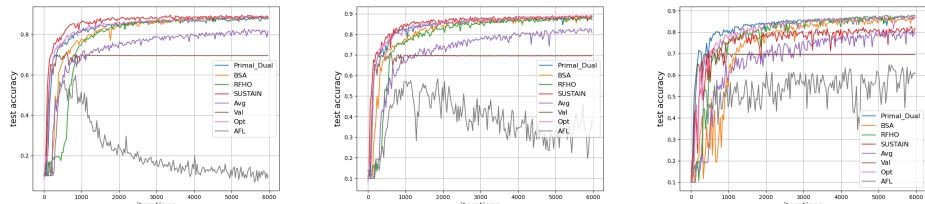

Figure 4: Test accuracy of training LeNet 5 on the Fashion-MNIST dataset. The left curve shows the result when the active probability is 1; the middle curve shows the result when the active probability is 0.9, and the right curve shows the result with 0.5 active probability.

## 6 CONCLUSION

In this paper, we proposed a primal-dual-based method for solving a bi-level optimization problem based on a federated learning tasking (local coefficient learning). We give a theoretical analysis that shows the convergence of the proposed algorithm. Though the analysis shows it needs more iterations for the algorithm to converge to an $\epsilon$-stationary point, it works well with a pretty small number of local steps in both toy case and neural network training. Other than that convergence rate can be improved (perhaps it should be in the order of $O(1/\sqrt{T})$ instead of $O(1/\sqrt{T} + 1/\sqrt{K})$), the initialization of dual variable affects the speed for convergence, which we leave as the future work.

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

## A   CONVERGENCE OF INNER LOOP

For simplicity, in this section we simplified the inner loop problem as the following:

$$\min_w g(w)$$
$$s.t. \ \nabla h(w) = 0.$$

Besides, the algorithm for solving the inner loop problem can be simplified as

$$w_{t+1} = w_t - \eta_w \left( \tilde{\nabla} g(w_t) + \tilde{\nabla}^2 h(w_t) \lambda_t + \Gamma \tilde{\nabla} h(w_t) \right) = w_t - \eta_w \tilde{\nabla} L(w_t, \lambda_t)$$

$$\lambda_{t+1} = \Pi_\Lambda \left( \lambda_t + \eta_\lambda \tilde{\nabla} h(w_t) \right)$$

Furthermore, the assumptions are as the following:

**(A1)** $g$ and $h$ are differentiable strongly convex function with modular $\mu$, $L_1$-Lipschitz gradient and lower bounded by $\underline{f}$.

**(A2)** $h$ has $L_2$-Lipschitz Hessian.

**(A3)** $\Lambda = \{\lambda | \|\lambda\| \le D_\lambda\}$, where $D \ge \|\nabla^2 h(w^*)^{-1} \nabla g(w^*)\|$ and $w^* = \arg\min h(w)$.

**(A4)** Given $w_t$, $\tilde{\nabla} g(w_t)$, $\tilde{\nabla}^2 h(w_t)$, $\tilde{\nabla} h(w_t)$ and $\hat{\nabla} h(w_t)$ are independent to each other. Besides, all of them are unbiased estimators with bounded variance with respect to mean value and a bounded constant $\sigma$, i.e.

$$\mathbb{E}\left[ \tilde{\nabla} g(w_t) \mid w_t \right] = \nabla g(w_t), \qquad \mathbb{E}\left[ \left\| \tilde{\nabla} g(w_t) - \nabla g(w_t) \right\|^2 \mid w_t \right] \le p\|\nabla g(w_t)\|^2 + \sigma^2;$$

$$\mathbb{E}\left[ \tilde{\nabla}^2 h(w_t) \mid w_t \right] = \nabla^2 h(w_t), \quad \mathbb{E}\left[ \left\| \tilde{\nabla}^2 h(w_t) - \nabla^2 h(w_t) \right\|_F^2 \mid w_t \right] \le p\|\nabla^2 h(w_t)\|^2 + \sigma^2;$$

$$\mathbb{E}\left[ \tilde{\nabla} h(w_t) \mid w_t \right] = \nabla h(w_t), \qquad \mathbb{E}\left[ \left\| \tilde{\nabla} h(w_t) - \nabla h(w_t) \right\|^2 \mid w_t \right] \le p\|\nabla h(w_t)\|^2 + \sigma^2.$$

Thus, it is easy to show $\mathbb{E}\left[ \tilde{\nabla}_w L(w_t, \lambda_t) \mid w_t, \lambda_t \right] = \nabla_w L(w_t, \lambda_t)$, and $\mathbb{E}\left[ \left\| \tilde{\nabla}_w L(w_t, \lambda_t) - \nabla_w L(w_t, \lambda_t) \right\|^2 \mid w_t, \lambda_t \right] \le (1 + \Gamma^2 + D_\lambda^2)(p\|\nabla_w L(w_t, \lambda_t)\|^2 + \sigma^2)$.

First, we give some notations that will be used in this section.

**Definition 1.** *Let* $L(w, \lambda) = g(w) + \lambda^\top \nabla h(w) + \Gamma h(w)$, $d(\lambda) = \min_w L(w, \lambda)$, $w^*(\lambda) = \arg\min_w L(w, \lambda)$, $\lambda_t^+ = \Pi_\Lambda(\lambda_t + \eta_\lambda \nabla h(w_t)))$ *and* $L_L = (1 + \Gamma)L_1 + D_\lambda L_2$.

*Define potential function* $\Phi(w_t, \lambda_t) = L(w_t, \lambda_t) - 2d(\lambda_t)$.

**Lemma 4** (Descent of Lagrange function). *For the function L, it holds that*

$$\mathbb{E}\left[ L(w_{t+1}, \lambda_{t+1}) - L(w_t, \lambda_t) \right]$$
$$\le \mathbb{E}\left[ C_0(\eta_w)\|\nabla_w L(w_t, \lambda_t)\|^2 + \frac{L_L \eta_w^2 (1 + \Gamma^2 + D_\lambda^2)\sigma^2}{2} + (\lambda_{t+1} - \lambda_t)^\top \nabla h(w_{t+1}) \right],$$

*where* $C_0(\eta_w) = \frac{L_L \eta_w^2}{2} + \frac{L_L \eta_w^2 p(1 + \Gamma^2 + D_\lambda^2)}{2} - \eta_w$.

*Proof.* Because $L(w, \lambda)$ has $L_L$ Lipschitz gradient, it holds that

$$L(w_{t+1}, \lambda_t) \leq L(w_t, \lambda_t) + \langle \nabla_w L(w_t, \lambda_t), w_{t+1} - w_t \rangle + \frac{L_L}{2} \|w_{t+1} - w_t\|^2.$$

By taking expectations with respect to $w_t$ and $\lambda_t$ on both side of the above inequality, we obtain that

$$\mathbb{E}\left[L(w_{t+1}, \lambda_t) - L(w_t, \lambda_t) \mid w_t, \lambda_t\right]$$

$$\leq \mathbb{E}\left[\langle \nabla_w L(w_t, \lambda_t), w_{t+1} - w_t \rangle + \frac{L_L}{2} \|w_{t+1} - w_t\|^2 \mid w_t, \lambda_t\right]$$

$$= \mathbb{E}\left[\left\langle \nabla_w L(w_t, \lambda_t), -\eta_w \tilde{\nabla}_w L(w_t, \lambda_t) \right\rangle + \frac{L_L \eta_w^2}{2} \|\nabla_w L(w_t, \lambda_t)\|^2 \right.$$

$$\left. + \frac{L_L \eta_w^2}{2} \|\tilde{\nabla}_w L(w_t, \lambda_t) - \nabla_w L(w_t, \lambda_t)\|^2 \mid w_t, \lambda_t\right]$$

$$= \mathbb{E}\left[\left(\frac{L_L \eta_w^2}{2} + \frac{L_L \eta_w^2 p(1 + \Gamma^2 + D_\lambda^2)}{2} - \eta_w\right) \|\nabla_w L(w_t, \lambda_t)\|^2 + \frac{L_L \eta_w^2 (1 + \Gamma^2 + D_\lambda^2)\sigma^2}{2} \mid w_t, \lambda_t\right].$$

(6)

Meanwhile, it holds that

$$\mathbb{E}\left[L(w_{t+1}, \lambda_{t+1}) - L(w_{t+1}, \lambda_t) \mid w_t, \lambda_t\right] = \mathbb{E}\left[(\lambda_{t+1} - \lambda_t)^\top \nabla h(w_{t+1}) | w_t, \lambda_t\right] \qquad (7)$$

Combining (6), (7) and taking expectation on the conditional expectation, we can obtain desired result. $\square$

**Lemma 5.** *Suppose* $\Gamma\mu > D_\lambda L_2 + L_1$*, it holds that*

$$\|w^*(\lambda_1) - w^*(\lambda_2)\| \leq \beta_1 \|\lambda_1 - \lambda_2\|$$

*for all* $\lambda_1, \lambda_2 \in \Lambda$*, where* $\beta_1 = \frac{L_1}{\Gamma\mu - D_\lambda L_2 - L_1}$*.*

*Proof.* Note that $L(w, \lambda)$ is a strongly convex function with respect to the $w$ in the domain $\lambda \in \Lambda$ with the modular $(1 + \Gamma)\mu - D_\lambda L_2$. Thus, it holds that

$$L(w^*(\lambda_1), \lambda_2) - L(w^*(\lambda_2), \lambda_2) \geq \frac{\Gamma\mu - D_\lambda L_2 - L_1}{2} \|w^*(\lambda_1) - w^*(\lambda_2)\|^2.$$

On the other hand, we have

$$L(w^*(\lambda_1), \lambda_2) - L(w^*(\lambda_2).\lambda_2)$$
$$= L(w^*(\lambda_1), \lambda_2) - L(w^*(\lambda_1), \lambda_1) + L(w^*(\lambda_1), \lambda_1) - L(w^*(\lambda_2), \lambda_1) + L(w^*(\lambda_2), \lambda_1) - L(w^*(\lambda_2), \lambda_2)$$
$$\leq (\lambda_1 - \lambda_2)^\top (\nabla h(w^*(\lambda_2)) - \nabla h(w^*(\lambda_1))) - \frac{\Gamma\mu - D_\lambda L_2 - L_1}{2} \|w^*(\lambda_1) - w^*(\lambda_2)\|^2.$$

Thus, by combining the above two inequalities, with Cauchy-Schwarz inequality, we can obtain

$$(\Gamma\mu - D_\lambda L_2 - L_1)\|w^*(\lambda_1) - w^*(\lambda_2)\|^2 \leq \|\lambda_1 - \lambda_2\|\|\nabla h(w^*(\lambda_1)) - \nabla h(w^*(\lambda_2))\|$$
$$\leq L_1 \|\lambda_1 - \lambda_2\|\|w^*(\lambda_1) - w^*(\lambda_2)\|.$$

Hence, we get the desired result. $\square$

**Lemma 6** (Ascent of dual function)**.** *It holds that*

$$\mathbb{E}\left[d(\lambda_{t+1}) - d(\lambda_t)\right] \geq \mathbb{E}\left[\langle \lambda_{t+1} - \lambda_t, \nabla h(w^*(\lambda_t))\rangle - \frac{L_d}{2} \|\lambda_{t+1} - \lambda_t\|^2\right].$$

*Proof.* It can be calculated by the implicit function theorem that

$$\nabla_\lambda d(\lambda) = \nabla h(w^*(\lambda)).$$

Thus, we can obtain that for all $\lambda_1, \lambda_2 \in \Lambda$

$$\|\nabla d(\lambda_1) - \nabla d(\lambda_2)\| = \|\nabla h(w^*(\lambda_1)) - \nabla h(w^*(\lambda_2))\| \leq L_1 \beta_1 \|\lambda_1 - \lambda_2\|.$$

Therefore, $d(\lambda)$ is a differentiable function with $L_d = L_1\beta_1$-Lipschitz gradient.

With the definition of $d(\lambda)$, we have

$$\mathbb{E}\left[d(\lambda_{t+1}) - d(\lambda_t)\right] \geq \mathbb{E}\left[\langle\lambda_{t+1} - \lambda_t, \nabla h(w^*(\lambda_t))\rangle - \frac{L_d}{2}\|\lambda_{t+1} - \lambda_t\|^2\right]$$

$\square$

**Lemma 7.** *With the strongly convexity of $L$ and $h$, it holds that*

$$\|w^*(\lambda_t) - w^*\| \leq \frac{1}{\mu}\|\nabla h(w^*(\lambda))\|,$$

*and*

$$\|w_t - w^*(\lambda_t)\| \leq \frac{1}{\Gamma\mu - D_\lambda L_2 - L_1}\|\nabla_w L(w_t, \lambda_t)\|$$

*Proof.* Because of the $\mu$-strongly convexity of $h$ the inequality $\mu\|w_1 - w_2\| \leq \|\nabla h(w_1) - \nabla h(w_2)\|$ holds for all $w_1, w_2$.

With the $\nabla h(w^*) = 0$, we get the result.

Similar to $h$, because of $\Gamma\mu - D_\lambda L_2 - L_1$-strongly convexity of $L$, we can prove the second inequality. $\square$

**Lemma 8** (Descent of one step local SGD). *It holds that*

$$\mathbb{E}\|w_t - w_{t+1}\|^2 \leq \mathbb{E}\left[\eta_w^2(1 + p)\|\nabla_w L(w_t, \lambda_t)\|^2 + \eta_w^2\sigma^2\right]$$

*Proof.*

$$\mathbb{E}\|w_t - w_{t+1}\|^2 = \mathbb{E}\left[\eta_w^2\|\tilde{\nabla}_w L(w_t, \lambda_t)\|^2\right] \leq \mathbb{E}\left[\eta_w^2(1 + p)\|\nabla_w L(w_t, \lambda_t)\|^2 + \eta_w^2\sigma^2\right]$$

$\square$

**Lemma 9.** $d(\lambda)$ *is a $\frac{\mu^2}{L_L}$-strongly concave funtion. We define $\mu_d = \frac{\mu^2}{L_L}$.*

*Proof.* Let $\gamma_0$ be the largest eigenvalue of $\nabla^2 d(\lambda)$, $\gamma_1$ be the largest eigenvalue of $\frac{\partial w^*(\lambda)}{\partial\lambda}$. Then, it holds that $\gamma \leq \gamma_1\mu$.

Meanwhile, $\frac{\partial w^*(\lambda)}{\partial\lambda} = -\nabla_w^2 L(w^*(\lambda), \lambda)^{-1}\nabla_w^2 h(w^*(\lambda))$. Thus, $\gamma_1 \leq -\frac{\mu}{L_L}$.

Therefore, $d(\lambda)$ is a $\frac{\mu^2}{L_L}$-strongly concave funtion. $\square$

**Lemma 10.** *It holds that*

$$\mathbb{E}\|\lambda_{t+1} - \lambda_t\|^2 \geq \frac{\mu_d^2\eta_\lambda^2}{16}\mathbb{E}\|\lambda_t - \lambda^*\|^2 - \frac{\mu_d\eta_\lambda}{4}\left(\eta_\lambda^2\sigma^2 + \left(2\eta_\lambda^2(1 + p)L_1^2 + \frac{L_1^2\eta_\lambda}{\mu_d}\right)\mathbb{E}\|w_t - w^*(\lambda_t)\|^2\right)$$

*Proof.* With the update rule of $\lambda_{t+1}$, it holds that

$$\begin{aligned}
&\mathbb{E}\|\lambda_{t+1} - \lambda^*\|^2\\
&= \mathbb{E}\|\Pi_\Lambda(\lambda_t + \eta_\lambda\nabla h(w_t)) - \lambda^*\|^2\\
&\leq \mathbb{E}\|\lambda_t + \eta_\lambda\tilde{\nabla h}(w_t) - \lambda^*\|^2\\
&= \mathbb{E}\left[\|\lambda_t - \lambda^*\|^2 + 2\eta_\lambda\langle\lambda_t - \lambda^*, \tilde{\nabla}h(w_t)\rangle + \eta_\lambda^2\|\tilde{\nabla}h(w_t)\|^2\right]\\
&= \mathbb{E}\left[\|\lambda_t - \lambda^*\|^2 + 2\eta_\lambda\langle\lambda_t - \lambda^*, \nabla h(w^*(\lambda_t))\rangle\right.\\
&\quad\left. + 2\eta_\lambda\langle\lambda_t - \lambda^*, \nabla h(w_t) - \nabla h(w^*(\lambda_t))\rangle + \eta_\lambda^2\|\tilde{\nabla}h(w_t)\|^2\right],
\end{aligned}$$

(8)

where the last equality is because $\tilde{\nabla}h(w_t)$ is an unbiased estimator.

Meanwhile, because $d(\lambda)$ is a strongly concave function, it holds that

$$\eta_\lambda \langle \lambda_t - \lambda^*, \nabla h(w^*(\lambda_t)) \rangle \leq -\eta_\lambda \mu_d \|\lambda_t - \lambda^*\|^2 \tag{9}$$

Further, it holds that

$$
\begin{aligned}
\mathbb{E}|\tilde{\nabla}h(w_t)\|^2 &= \mathbb{E}\|\tilde{\nabla}h(w_t) + \nabla h(w_t) - \nabla h(w_t)\|^2 \\
&\leq \sigma^2 + \mathbb{E}(1+p)\|\nabla h(w_t)\|^2 \\
&\leq \sigma^2 + \mathbb{E}2(1+p)\|\nabla h(w^*(\lambda_t)))\|^2 + 2(1+p)\|\nabla h(w^*(\lambda_t) - \nabla h(w_t)\|^2 \\
&\leq \sigma^2 + \mathbb{E}2(1+p)\|\nabla h(w^*(\lambda_t)))\|^2 + 2(1+p)L_1^2\|w^*(\lambda_t - w_t\|^2 \\
&\leq \sigma^2 + \mathbb{E}2(1+p)L_d^2\|\lambda_t - \lambda^*\|^2 + 2(1+p)L_1^2\|w^*(\lambda_t) - w_t\|^2
\end{aligned}
\tag{10}
$$

For $\langle \lambda_t - \lambda^*, \nabla h(w_t) - \nabla h(w^*(\lambda_t)) \rangle$, it holds that

$$
\begin{aligned}
\langle \lambda_t - \lambda^*, \nabla h(w_t) - \nabla h(w^*(\lambda_t)) \rangle &\leq \frac{\mu_d}{2}\|\lambda_t - \lambda^*\|^2 + \frac{1}{2\mu_d}\|\nabla h(w_t) - \nabla h(w^*(\lambda_t)\|^2 \\
&\leq \frac{\mu_d}{2}\|\lambda_t - \lambda^*\|^2 + \frac{L_1^2}{2\mu_d}\|w_t - w^*(\lambda_t)\|^2
\end{aligned}
\tag{11}
$$

Combining (8), (9), (10) and (11), it holds that

$$\mathbb{E}\|\lambda_{t+1} - \lambda^*\|^2$$

$$\leq (1 - \mu_d\eta_\lambda + 2(1+p)L_d^2\eta_\lambda^2)\mathbb{E}\|\lambda_t - \lambda^*\|^2 + \eta^2\sigma^2 + \left(2\eta_\lambda^2(1+p)L_1^2 + \frac{L_1^2\eta_\lambda}{\mu_d}\right)\mathbb{E}\|w_t - w^*(\lambda_t)\|^2$$

When $\eta_\lambda \leq \frac{\mu_d}{4(1+p)L_d^2}$, it holds that

$$\mathbb{E}\|\lambda_{t+1} - \lambda^*\|^2 \leq (1 - \frac{\mu_d\eta_\lambda}{2})\mathbb{E}\|\lambda^* - \lambda_t\|^2 + \eta_\lambda^2\sigma^2 + \left(2\eta_\lambda^2(1+p)L_1^2 + \frac{L_1^2\eta_\lambda}{\mu_d}\right)\mathbb{E}\|w_t - w^*(\lambda_t)\|^2. \tag{12}$$

It holds that

$$
\begin{aligned}
\mathbb{E}\|\lambda_{t+1} - \lambda_t\|^2 &= \mathbb{E}\|\lambda_{t+1} - \lambda^* + \lambda^* - \lambda_t\|^2 \\
&= \mathbb{E}\left[\|\lambda_{t+1} - \lambda^*\|^2 + \|\lambda^* - \lambda_t\|^2 + 2\langle\lambda_{t+1} - \lambda^*, \lambda^* - \lambda_t\rangle\right] \\
&\geq \mathbb{E}\left[(1-\xi)\|\lambda_{t+1} - \lambda^*\|^2 + \left(1 - \frac{1}{\xi}\right)\|\lambda^* - \lambda_t\|^2\right], \forall \xi > 1
\end{aligned}
\tag{13}
$$

Let $\xi = 1 + \mu_d\eta_\lambda/4$, $(1-\xi)\left(1 - \frac{\mu_d\eta_\lambda}{2}\right) + \left(1 - \frac{1}{\xi}\right) \geq \mu_d^2\eta_\lambda^2/16$. combining (12) and (13), it holds that

$$\mathbb{E}\|\lambda_{t+1} - \lambda_t\|^2$$

$$\geq \left((1-\xi)\left(1 - \frac{\mu_d\eta_\lambda}{2}\right) + \left(1 - \frac{1}{\xi}\right)\right)\|\lambda^* - \lambda_t\|^2 + (1-\xi)\left(\eta_\lambda^2\sigma^2 + \left(2\eta_\lambda^2(1+p)L_1^2 + \frac{L_1^2\eta_\lambda}{\mu_d}\right)\mathbb{E}\|w_t - w^*(\lambda_t)\|^2\right)$$

$$\geq \frac{\mu_d^2\eta_\lambda^2}{16}\mathbb{E}\|\lambda_t - \lambda^*\|^2 - \frac{\mu_d\eta_\lambda}{4}\left(\eta_\lambda^2\sigma^2 + \left(\frac{2\eta_\lambda^2(1+p)L_1^2}{\Gamma\mu - D_\lambda L_2 - L_1} + \frac{L_1^2\eta_\lambda}{\mu_d\Gamma\mu - D_\lambda L_2 - L_1}\right)\mathbb{E}\|w_t - w^*(\lambda_t)\|^2\right)$$

Thus, we get the desired result. $\qquad\square$

Define a potential function $\Phi(w_t, \lambda_t) = L(w_t, \lambda_t) - 2d(\lambda_t)$. According to the definition we have $L(w_t, \lambda_t) > d(\lambda_t)$. Then, $\Phi(w_t, \lambda_t) \geq -d(\lambda_t) \geq -\min_w g(w) + \Gamma h(w) \geq -(1+\Gamma)\underline{f}$ for all $w_t$ and $\lambda_t \in \Lambda$.

**Lemma 11** (Descent of potential function). *It holds that*

$$\mathbb{E}\left[\Phi(w_{t+1}, \lambda_{t+1}) - \Phi(w_t, \lambda_t)\right]$$
$$\leq \mathbb{E}\left[C_1 \|\nabla_w L(w_t, \lambda_t)\|^2 + C_2 \|\lambda_t - \lambda_t^*\|^2\right] + C_3$$

*where*

$$C_1 = -\eta_w + \frac{L_L \eta_w^2}{2} + \frac{L_L \eta_w^2 p (1 + \Gamma^2 + D_\lambda^2)}{2} + 4L_1^2 \eta_\lambda \eta_w^2 (1 + p)$$
$$+ \frac{4L_1^2 \eta_\lambda}{(\Gamma\mu - D_\lambda L_2 - L_1)^2} + \frac{\mu_d}{16}\left(2\eta_\lambda^2(1+p)L_1^2 + \frac{L_1^2 \eta_\lambda}{\mu_d}\right)$$

$$C_2 = -\frac{\eta_\lambda \mu_d^2}{64}$$

$$C_3 = \frac{L_L \eta_w^2 (1 + \Gamma^2 + D_\lambda^2)\sigma^2}{2} + 2L_1^2 \eta_\lambda \eta_w^2 \sigma^2 + 2\eta_\lambda^2 \sigma^2 + \frac{\mu_d \eta_\lambda^2 \sigma^2}{16}.$$

*Proof.* With Lemma 4 and Lemma 6, it holds that

$$\mathbb{E}\left[\Phi(w_{t+1}, \lambda_{t+1}) - \Phi(w_t, \lambda_t)\right]$$
$$\leq \mathbb{E}\left[C_0(\eta_w)\|\nabla_w L(w_t, \lambda_t)\|^2 + (\lambda_{t+1} - \lambda_t)^\top \nabla h(w_{t+1}) - 2\langle \lambda_{t+1} - \lambda_t, \nabla h(w^*(\lambda_t))\rangle + L_d \|\lambda_{t+1} - \lambda_t\|^2\right]$$
$$+ \frac{L_L \eta_w^2 (1 + \Gamma^2 + D_\lambda^2)\sigma^2}{2}.$$

We deal with each term as follows. For the second term and the third term, it holds that

$$(\lambda_{t+1} - \lambda_t)^\top \nabla h(w_{t+1}) - 2\langle \lambda_{t+1} - \lambda_t, \nabla h(w^*(\lambda_t))\rangle$$
$$= 2\langle \lambda_{t+1} - \lambda_t, \nabla h(w_{t+1}) - \nabla h(w^*(\lambda_t))\rangle - (\lambda_{t+1} - \lambda_t)^\top \nabla h(w_{t+1})$$
$$\leq 2L_1 \|\lambda_{t+1} - \lambda_t\| \|w_{t+1} - w^*(\lambda_t)\| - \frac{1}{\eta_\lambda}\|\lambda_{t+1} - \lambda_t\|^2$$
$$\leq 2L_1 \|\lambda_{t+1} - \lambda_t\| \left(\|w_t - w^*(\lambda_t))\| + \|w_{t+1} - w_t\|\right) - \frac{1}{\eta_\lambda}\|\lambda_{t+1} - \lambda_t\|^2$$
$$\leq \left(\frac{1}{2\eta_\lambda} - \frac{1}{\eta_\lambda}\right)\|\lambda_{t+1} - \lambda_t\|^2 + 4L_1^2 \eta_\lambda \|w_t - w^*(\lambda_t))\|^2 + 4L_1^2 \eta_\lambda \|w_{t+1} - w_t\|^2$$

By taking the expectation on the both side of inequality, it holds that

$$\mathbb{E}\left[(\lambda_{t+1} - \lambda_t)^\top \nabla h(w_{t+1}) - 2\langle \lambda_{t+1} - \lambda_t, \nabla h(w^*(\lambda_t))\rangle\right]$$
$$\leq \mathbb{E}\left[-\frac{1}{2\eta_\lambda}\|\lambda_{t+1} - \lambda_t\|^2 + 4L_1^2 \eta_\lambda \|w_t - w^*(\lambda_t))\|^2 + 4L_1^2 \eta_\lambda \|w_{t+1} - w_t\|^2\right]$$
$$\leq \mathbb{E}\left[-\frac{1}{2\eta_\lambda}\|\lambda_{t+1} - \lambda_t\|^2 + 4L_1^2 \eta_\lambda \eta_w^2(1+p)\|\nabla_w L(w_t, \lambda_t)\|^2 + \frac{4L_1^2 \eta_\lambda}{(\Gamma\mu - D_\lambda L_2 - L_1)^2}\|\nabla_w L(w_t, \lambda_t)^2\|^2\right] + 2L_1^2 \eta_\lambda \eta_w^2 \sigma^2.$$

Meanwhile, with Lemma 10 and Lemma 7 it holds that

$$\mathbb{E}\|\lambda_{t+1} - \lambda_t\|^2$$
$$\geq \frac{\mu_d^2 \eta_\lambda^2}{16}\mathbb{E}\|\lambda_t - \lambda^*\|^2 - \frac{\mu_d \eta_\lambda}{4}\left(\eta_\lambda^2 \sigma^2 + \left(2\eta_\lambda^2(1+p)L_1^2 + \frac{L_1^2 \eta_\lambda}{\mu_d}\right)\mathbb{E}\|w_t - w^*(\lambda_t)\|^2\right)$$
$$\geq \frac{\mu_d^2 \eta_\lambda^2}{16}\mathbb{E}\|\lambda_t - \lambda^*\|^2 - \frac{\mu_d \eta_\lambda}{4}\left(\eta_\lambda^2 \sigma^2 + \left(2\eta_\lambda^2(1+p)L_1^2 + \frac{L_1^2 \eta_\lambda}{\mu_d}\right)\mathbb{E}\|\nabla_w L(w_t, \lambda_t)\|^2\right)$$

Further, when $\eta_\lambda \leq 1/(4L_d)$, it holds that $-\frac{1}{2\eta_\lambda}\|\lambda_{t+1} - \lambda_t\|^2 + L_d\|\lambda_{t+1} - \lambda_t\|^2 \leq \frac{1}{4\eta_\lambda}$.

Thus, it holds that

$$\mathbb{E}\left[\Phi(w_{t+1}, \lambda_{t+1}) - \Phi(w_t, \lambda_t)\right]$$
$$\leq \mathbb{E}\left[C_0(\eta_w)\|\nabla_w L(w_t, \lambda_t)\|^2 + (\lambda_{t+1} - \lambda_t)^\top \nabla h(w_{t+1}) - 2\langle \lambda_{t+1} - \lambda_t, \nabla h(w^*(\lambda_t))\rangle + L_d \|\lambda_{t+1} - \lambda_t\|^2\right]$$
$$+ \frac{L_L \eta_w^2 (1 + \Gamma^2 + D_\lambda^2)\sigma^2}{2}$$
$$\leq \mathbb{E}\left[C_1\|\nabla_w L(w_t, \lambda_t)\|^2 + C_2\|\lambda_t - \lambda_t^*\|^2 + C_3\right],$$

where

$$C_1 = -\eta_w + \frac{L_L \eta_w^2}{2} + \frac{L_L \eta_w^2 p(1 + \Gamma^2 + D_\lambda^2)}{2} + 4L_1^2 \eta_\lambda \eta_w^2 (1 + p)$$
$$+ \frac{4L_1^2 \eta_\lambda}{(\Gamma\mu - D_\lambda L_2 - L_1)^2} + \frac{\mu_d}{16}\left(2\eta_\lambda^2(1 + p)L_1^2 + \frac{L_1^2 \eta_\lambda}{\mu_d}\right)$$
$$C_2 = -\frac{\eta_\lambda \mu_d^2}{64}$$
$$C_3 = \frac{L_L \eta_w^2 (1 + \Gamma^2 + D_\lambda^2)\sigma^2}{2} + 2L_1^2 \eta_\lambda \eta_w^2 \sigma^2 + 2\eta_\lambda^2 \sigma^2 + \frac{\mu_d \eta_\lambda^2 \sigma^2}{16}.$$

$\square$

*Proof.* **Proof of the Inner Convergence**

By the definition of $\Phi$, it holds that $\Phi(w, \lambda) \geq -d(\lambda) \geq \underline{f}$.

Thus, with Lemma 11, by summing up $T$ terms, it holds that

$$\frac{1}{K}\sum_{t=1}^{K}\mathbb{E}\frac{\eta_w}{2}\|\nabla_w L(w_t, \lambda_t)\|^2 + \frac{\eta_\lambda L_L^2}{8\mu^4}\|\lambda_t - \lambda^*\|^2 \leq \frac{\Phi(w_1, \lambda_1) - \underline{f}}{K} + C_3$$

Then, let $\eta_w = \Theta(1/\sqrt{K})$ and $\eta_\lambda = \Theta(1/\sqrt{K})$, it holds that

$$\frac{1}{K}\mathbb{E}\sum_{t=1}^{K}\|\lambda_t - \lambda^*\|^2 = O(1/\sqrt{K})$$

$$\frac{1}{K}\mathbb{E}\sum_{t=1}^{K}\|w_t - w^*\|^2 = O(1/\sqrt{K})$$

On the other hand, it holds that

$$\|\lambda_t^\top \nabla_w f_i(w_t) - \lambda^{*\top}\nabla_w f_i(w^*)\|^2 \leq 2\|\nabla_w f_i(w^*)\|^2\|\lambda_t - \lambda^*\|^2 + 2\|\lambda_t\|^2\|\nabla f_i(w_t) - \nabla f_i(w^*)\|^2$$
$$\leq 2D_w^2\|\lambda_t - \lambda^*\|^2 + 2D_\lambda^2 L_1^2\|w_t - w^*\|^2$$

Therefore, it holds that

$$\frac{1}{K}\mathbb{E}\sum_{t=1}^{K}\|\lambda_t^\top \nabla_w f_i(w_t) - \lambda^{*\top}\nabla_w f_i(w^*)\|^2 \leq \frac{2}{K}\mathbb{E}\sum_{t=1}^{K}D_w^2\|\lambda_t - \lambda^*\|^2 + D_\lambda^2 L_1^2\|w_t - w^*\|^2 = O(1/\sqrt{K})$$

$\square$

## B  PROOF OF OUTER LOOP CONVERGENCE

In this section, we give the proof of the outer loop convergence.

**Lemma 12.** *Suppose **(A1)-(A4)** holds. Then, for all $x_1, x_2 \in \mathcal{X}$, it holds that*

$$\|w^*(x_1) - w^*(x_2)\| \leq \frac{\sqrt{N}D_w \mu + \sqrt{N}D_w L_1}{\mu^2}\|x_1 - x_2\|$$

*Proof.* Because $\sum_{i=1}^{N} x_2^{(i)} f_i(w)$ is $\mu$ is a strongly convex function, it holds that

$$\sum_{i=1}^{N} x_2^{(i)}(f_i(w^*(x_1)) - f_i(w^*(x_2))) \geq \frac{\mu}{2}\|w^*(x_1) - w^*(x_2)\|^2.$$

On the other hand, we have

$$\sum_{i=1}^{N} x_2^{(i)} (f_i(w^*(x_1)) - f_i(w^*(x_2)))$$

$$= \sum_{i=1}^{N} x_2^{(i)} f_i(w^*(x_1)) - \sum_{i=1}^{N} x_1^{(i)} f_i(w^*(x_1)) + \sum_{i=1}^{N} x_1^{(i)} f_i(w^*(x_1))$$

$$- \sum_{i=1}^{N} x_1^{(i)} f_i(w^*(x_2)) + \sum_{i=1}^{N} x_1^{(i)} f_i(w^*(x_2)) - \sum_{i=1}^{N} x_2^{(2)} f_i(w^*(x_2))$$

$$\leq \sum_{i=1}^{N} (x_1^{(i)} - x_2^{(i)})(f_i(w^*(x_2)) - f_i(w^*(x_1))) - \frac{\mu}{2} \|w^*(x_2) - w^*(x_1)\|^2.$$

Meanwhile, we have

$$|f_i(w^*(x_1)) - f_i(w^*(x_2))| \leq \max(\|\nabla_w f(w^*(x_1))\|, \|\nabla_w f(w^*(x_2))\|)\|w^*(x_1) - w^*(x_2)\| + \frac{L_1}{2}\|w^*(x_2) - w^*(x_1)\|^2$$

With strongly convexity, it holds that

$$\|w^*(x_2) - w^*(x_1)\| \leq \frac{1}{\mu}\|\nabla f(w^*(x_1)) - \nabla f(w^*(x_2))\| \leq \frac{2D_w}{\mu}.$$

Thus, it holds that

$$f_i(w^*(x_2)) - f_i(w^*(x_1)) \leq D_w(1 + L_1/\mu)\|w^*(x_2) - w^*(x_1)\|.$$

Combining the above inequalities, it holds that

$$\mu\|w^*(x_1) - w^*(x_2)\|^2 \leq \|x_1 - x_2\|_1 (max|f_i(w^*(x_1)) - f_i(w^*(x_2))|) \leq \sqrt{N} D_w(1 + L_1/\mu)\|w^*(x_1) - w^*(x_2)\|$$

$$\square$$

**Lemma 13.** *Suppose* **(A1)-(A4)** *holds, then* $f_0(w^*(x))$ *has Lipschitz gradient with Lipschitz constant* $\left(\frac{D_w^2}{\mu^2} + \frac{2D_w L_1}{\mu}\right) \frac{\sqrt{N}D_w\mu + \sqrt{N}D_w L_1}{\mu^2}.$

*Proof.*

$$\frac{\partial f_0(w^*(x))}{\partial x^{(i)}} = -\nabla_w f_0(w^*(x))^\top \left(\sum_{j=1}^{N} x^{(j)}\nabla_w^2 \nabla f_j(w^*(x))\right)^{-1} \nabla_w f_i(w^*(x))$$

By the smoothness and strongly convexity it holds that

$$\left\|\left(\sum_{j=1}^{N} x_1^{(j)}\nabla_w^2 \nabla f_j(w^*(x_1))\right)^{-1} - \left(\sum_{j=1}^{N} x_2^{(j)}\nabla_w^2 \nabla f_j(w^*(x_2))\right)^{-1}\right\|$$

$$\leq \frac{1}{\mu^2}\left\|\sum_{j=1}^{N} x_1^{(j)}\nabla_w^2 \nabla f_j(w^*(x_1)) - \sum_{j=1}^{N} x_2^{(j)}\nabla_w^2 \nabla f_j(w^*(x_2))\right\|$$

$$\leq \frac{1}{\mu^2} \max_j \|\nabla_w^2 \nabla f_j(w^*(x_1)) - \nabla_w^2 \nabla f_j(w^*(x_2))\|$$

$$\leq \frac{L_2}{\mu^2}\|w^*(x_1) - w^*(x_2)\|,$$

where the first inequality is due to $\|A(A^{-1} - B^{-1})B\| = \|A - B\|$ for all invertible matrices $A, B$. Then, it holds that

$$
\left\| \left( \sum_{j=1}^{N} x_1^{(j)} \nabla_w^2 \nabla f_j(w^*(x_1)) \right)^{-1} \nabla_w f_i(w^*(x_1)) - \left( \sum_{j=1}^{N} x_2^{(j)} \nabla_w^2 \nabla f_j(w^*(x_2)) \right)^{-1} \nabla_w f_i(w^*(x_2)) \right\|
$$

$$
\leq \left\| \left( \left( \sum_{j=1}^{N} x_1^{(j)} \nabla_w^2 \nabla f_j(w^*(x_1)) \right)^{-1} - \left( \sum_{j=1}^{N} x_2^{(j)} \nabla_w^2 \nabla f_j(w^*(x_2)) \right)^{-1} \right) \nabla_w f_i(w^*(x_1)) \right\|
$$

$$
+ \left\| \left( \sum_{j=1}^{N} x_2^{(j)} \nabla_w^2 \nabla f_j(w^*(x_2)) \right)^{-1} (\nabla f_i(w^*(x_1)) - \nabla f_i(w^*(x_2))) \right\|
$$

$$
\leq \frac{D_w}{\mu^2} \|w^*(x_1) - w^*(x_2)\| + \frac{L_1}{\mu} \|w^*(x_1) - w^*(x_2)\|.
$$

Thus, combining with the definition of gradient, it holds that

$$
\left| \frac{\partial f_0(w^*(x_1))}{\partial x^{(i)}} - \frac{\partial f_0(w^*(x_2))}{\partial x^{(i)}} \right|
$$

$$
\leq \|\nabla_w f_0(w^*(x_1))\| \left\| \left( \sum_{j=1}^{N} x_1^{(j)} \nabla_w^2 \nabla f_j(w^*(x_1)) \right)^{-1} \nabla_w f_i(w^*(x_1)) - \left( \sum_{j=1}^{N} x_2^{(j)} \nabla_w^2 \nabla f_j(w^*(x_2)) \right)^{-1} \nabla_w f_i(w^*(x_2)) \right\|
$$

$$
+ \|\nabla_w f_0(w^*(x_1)) - \nabla_w f_0(w^*(x_2))\| \left\| \left( \sum_{j=1}^{N} x_2^{(j)} \nabla_w^2 \nabla f_j(w^*(x_2)) \right)^{-1} \nabla_w f_i(w^*(x_2)) \right\|
$$

$$
\leq D_w (\frac{D_w}{\mu^2} \|w^*(x_1) - w^*(x_2)\| + \frac{L_1}{\mu} \|w^*(x_1) - w^*(x_2)\|) + \frac{D_w L_1}{\mu} \|w^*(x_1) - w^*(x_2)\|
$$

Therefore, combine with Lemma 12, we can obtain the result. $\qquad\square$

**Lemma 14.** *Suppose function $f(x)$ is lower bounded by $\underline{f}$ with L-Lipshitz gradient. $g(x)$ is an unbiased gradient estimator of $\nabla f(x)$ satisfying that expected norm of g(x) are bounded by $G$ in domain $\mathcal{X}$ for function $f$. Then with update rule $x_{t+1} = \Pi_{\mathcal{X}}(x_t - \eta_x(g(x_t) + \xi_t))$, where $\eta_x = \Theta(1/\sqrt{T})$, $\mathcal{X}$ is a convex set and $\mathbb{E}\|\xi_t\|^2 \leq \epsilon^2$. By defining $\hat{x} = \arg\min_{y \in \mathcal{X}}(f(y) + \frac{\rho}{2}\|y - x\|^2)$ and $\bar{\nabla}_\rho f(x) = \rho(x - \hat{x})$, where $\rho = 2L$, then it holds that*

$$
\frac{1}{T} \mathbb{E} \sum_{t=1}^{T} \|\bar{\nabla}_\rho f(x_t)\|^2 = O(1/\sqrt{T} + \epsilon^2).
$$

*Proof.* It holds that

$$
\mathbb{E}\left( f(\hat{x}_{t+1} + \frac{\rho}{2}\|x_{t+1} - \hat{x}_{t+1}\|^2 \right)
$$

$$
\leq \mathbb{E}\left( f(\hat{x}_t) + \frac{\rho}{2}\|x_{t+1} - \hat{x}_t\|^2 \right)
$$

$$
= \mathbb{E}\left( f(\hat{x}_t) + \frac{\rho}{2}\|\Pi_{\mathcal{X}}(x_t - \eta_x(g(x_t) + \xi_t)) - \hat{x}_t\|^2 \right)
$$

$$
\leq \mathbb{E}\left( f(\hat{x}_t) + \frac{\rho}{2}\|x_t - \eta_x(g(x_t) + \xi_t) - \hat{x}_t\|^2 \right)
$$

$$
= \mathbb{E}\left( f(\hat{x}_t) + \frac{\rho}{2}\|x_t - \hat{x}_t\|^2 + \rho\eta_x^2\|g(x_t) + \xi_t\|^2 - \eta_x\rho\langle x_t - \hat{x}_t, g(x_t)\rangle - \eta_x\rho\langle x_t - \hat{x}_t, \xi_t\rangle \right)
$$

$$
\leq \mathbb{E}\left( f(\hat{x}_t) + \frac{\rho}{2}\|x_t - \hat{x}_t\|^2 \right) + \rho\eta_x^2(G^2 + \epsilon^2) - \rho\eta_x\mathbb{E}\left( f(x_t) - f(\hat{x}_t) - \frac{L}{2}\|x_t - \hat{x}_t\|^2 \right)
$$

$$
+ \rho\eta_x\mathbb{E}\left( \frac{L}{2}\|x_t - \hat{x}_t\|^2 + \frac{1}{2L}\epsilon^2 \right)
$$

Then summing up the above inequality, it holds that

$$\rho\eta_x \sum_{t=1}^{T} \mathbb{E}\left(f(x_t) - f(\hat{x}_t) - L\|x_t - \hat{x}_t\|^2\right)$$

$$\leq \mathbb{E}\left[f(\hat{x}_1) + \frac{\rho}{2}\|x_1 - \hat{x}_1\|^2 - (f(\hat{x}_{T+1}) + \frac{\rho}{2}\|x_{T+1} - \hat{x}_{T+1}\|^2)\right] + T\rho\eta_x^2(G^2 + \epsilon^2) + \frac{T\rho\eta_x}{2}\epsilon^2$$

$$\leq f(\hat{x}_1) + \frac{\rho}{2}\|x_1 - \hat{x}_1\|^2 - \underline{f}) + T\rho\eta_x^2(G^2 + \epsilon^2) + \frac{T\rho\eta_x}{2L}\epsilon^2$$

On the other hand, because $\hat{x}_t = \arg\min_{y \in \mathcal{X}}(f(y) + \frac{\rho}{2}\|x_t - y\|^2)$ and function $f(y) + \frac{\rho}{2}\|x_t - y\|^2$ is strongly convex with modular $\rho - L$, it holds that

$$f(x_t) - f(\hat{x}_t) - L\|x_t - \hat{x}_t\|^2$$

$$= f(x_t) + \frac{\rho}{2}\|x_t - x_t\|^2 - (f(\hat{x}_t) + \frac{\rho}{2}\|x_t - \hat{x}_t\|^2) + \frac{\rho - L - 1}{2}\|x_t - \hat{x}_t\|^2$$

$$\geq \frac{2\rho - 3L}{2}\|x_t - \hat{x}_t\|^2 = \frac{L}{2}\|x_t - \hat{x}_t\|^2$$

Thus, combining the above inequalities, it holds that

$$\frac{\rho^2}{T}\sum_{t=1}^{T}\mathbb{E}\|x_t - \hat{x}_t\|^2$$

$$\leq \frac{2\rho^2}{TL}\sum_{t=1}^{T}\mathbb{E}f(x_t) - f(\hat{x}_t) - L\|x_t - \hat{x}_t\|^2$$

$$\leq \frac{\rho f(\hat{x}_1) + \frac{\rho^2}{2}\|x_1 - \hat{x}_1\|^2 - \rho\underline{f})}{T\eta_x} + \rho^2\eta_x(G^2 + \epsilon^2) + \frac{\rho^2}{2L}\epsilon^2.$$

Therefore, when $\eta_x = O(1/\sqrt{T})$, $\frac{1}{T}\mathbb{E}\sum_{t=1}^{T}\|\bar{\nabla}_\rho f(x_t)\|^2 = \frac{\rho^2}{T}\sum_{t=1}^{T}\mathbb{E}\|x_t - \hat{x}_t\|^2 = O(1/\sqrt{T} + \epsilon^2)$. $\qquad\square$

**Proof of Theorem 2.** With Lemma 13, we can obtain $f_0(w^*(x))$ has Lipschitz gradient on domain $\mathcal{X}$.

Define $\xi_t^{(i)} = \lambda_t^T \nabla_w f_i(w_{t,k}) - \lambda^*(x)^T \nabla_w f_i(w^*(x))$

When we use $\lambda_{t,k}$ and $w_{t,k}$ with random k, as it suggests in Theorem 1, the expected norm of $\xi_t^{(i}$ can be bounded by $O(1/\sqrt{K})$.

Let $x_{t+1/2}^{(i)} = x_t^{(i)} - \eta_x(g(x_t)^{(i)} + \lambda_t^T \nabla_w f_i(w_{t,k}) - \lambda^*(x_t)^T \nabla_w f_i(w^*(x_t)))$,

Then, it holds that

$$\mathbb{E}g(x_t)^{(i)} = \mathbb{E}\left[\frac{1}{\eta_x}(x_{t+1/2} - x_t) - \lambda_t^T \nabla_w f_i(w_{t,k}) + \lambda^*(x_t)^T \nabla_w f_i(w^*(x_t))\right] = \lambda^*(x_t)^T \nabla_w f_i(w^*(x_t))$$

and

$$\mathbb{E}\left((g(x_t)^{(i)})^2\right) \leq (1+p)\|\lambda^*(x_t)^T \nabla_w f_i(w^*(x_t)))\|^2 + \epsilon^2 + \sigma^2 \leq (1+p)D_\lambda^2 D_w^2 + \epsilon^2 + \sigma^2$$

Thus, together with Lemma 14, we can directly get the result. $\qquad\square$

## C   ADDITIONAL EXPERIMENTAL RESULTS ON MNIST AND FASHION MNIST WITH LENET5

We use 10 clients in this experiment. The first 5 clients contains i.i.d. 9000 samples, the last 5 clients contains 9000 samples with random label. The rest setting is the same as it in the main text. The results are shown in Figure 5, and Figure 6.

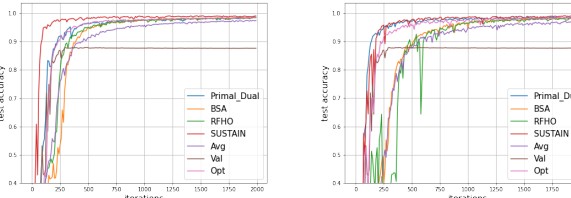

Figure 5: Test accuracy of training LeNet5 on MNIST dataset in iid case. The left curve shows the result when active probability is 1, and the right figure shows the result when active probability is 0.5.

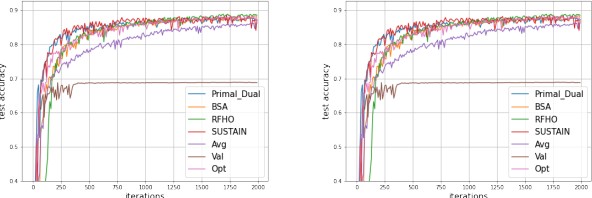

Figure 6: Test accuracy of training LeNet5 on Fashion-MNIST dataset in iid case. The left curve shows the result when active probability is 1, and the right figure shows the result when active probability is 0.5.

