# OpenReview forum: "Local Coefficient Optimization in Federated Learning"
_ICLR.cc/2023/Conference — Submitted to ICLR 2023_

### Official Review · Reviewer_KXgD · 2022-10-16

**Confidence:** 3
**Correctness:** 4
**Technical Novelty And Significance:** 2
**Empirical Novelty And Significance:** 2
**Recommendation:** 3

**Clarity, Quality, Novelty And Reproducibility:**

Clarity:
The mathematics and experimental results are very clear. The motivation of the paper is not clear. I think more words need to be added to discuss the motivation of the problem.

Quality:
The theoretical results and empirical results have high quality. The proof seems correct to me, though I did not check them all.


Novelty:
Given the current writing, it is hard for me to detect enough novelty. I do think the augmented Lagrangian term is a novel design. However, I did not see enough discussion on the technical challenges and novelty.

Reproducibility:
The reproducibility is good.


**Strength And Weaknesses:**

Strength:

1. The presentation is good and easy to follow.

2. The mathematics seems correct (though I did not check all the proof details).

3. The problem itself (i.e. equation 1) is meaningful.

4. Extensive experiments are conducted to support the theory.

---
Weakness:

Not well-motivated: I indeed believe problem eq (1) is a very meaningful problem. But I do not think there is enough motivation in this paper.

i) the original motivation is that one would like to train a model parameter that is good for some global task $f_0$. This makes sense but why not directly optimize for $f_0$? The reason is (my understanding) there is not enough data for optimization w.r.t. $f_0$ directly and therefore we need some pre-training using all local clients’ data. Motivation like this should be emphasized.

ii) under the same motivation, equation 1 is only one of the many ways to formulate the problem. It would be necessary to discuss alternative problem formulation as well. For example, why not first pretrain on local clients data using federated learning and then fine-tuning on few-shot data from task $f_0$?

iii) When I first read, it was not immediately unclear to me why the estimation of Hessian in the previous methods is hard to implement under this paper’s setting due to the large amount of data. My understanding is one cannot get an unbiased estimator of the Hessian. Issues like this should also be discussed in more depth because it motivates the paper.

2. The paper considers the case where all the $f_i$’s are strongly convex, which is the most simplified case. That is fine. But the paper did not explicitly discuss the unique technical challenges under this setting. It is hard to justify the technical novelty given the current writing. I do believe the augmented Lagrangian term is novel but it is hard to judge given the current writing. Besides, the algorithmic design section is also hard for me to spot what is the novelty. I think the authors should emphasize the technical challenges and novelty.

---
Questions:

1. Motivate problem formulation given by equation (1). Especially, what is the relation between this problem and meta-learning? In Meta-learning, one would like to find a model parameter that is good for (possibly unseen) downstream tasks. This is clearly very related with this problem considered in this paper.

2. In the paragraph below equation 5, the paper mentions that the term $\tilde{\nabla}^2 f_i(w_{t,k})$ is calculated with auto-differentiation. Explain why we need to calculate this term and why it is easier than the calculation of Hessian in previous approaches.

---
Minor comments/suggestions/typos:
1. Typo in Alg 1 line 7.


**Summary Of The Paper:**

This paper deals with a specific setting of Federated Learning where the goal is to find an optimal weight between all the local objectives so that the learned parameter can be optimal for some different global task. To do so, this paper first formulates the above setting as a bi-level optimization problem over the weight parameter and the model parameter. Then, to solve this bi-level optimization problem, the paper reformulates the bi-level problem into an optimization problem with equality constraint using convexity assumption. Then the paper introduces a primal-dual algorithm and gives convergence guarantee under the case where each local objective function is strongly convex. Since it is possible to encounter a constrained optimization problem with non-linear equality constraints, the paper proposes a new Lagrangian formulation with an augmented Lagrangian term. They shows that due to this new term, the primal-dual optimizer can converge to the KKT point of the original bi-level optimization problem.

**Summary Of The Review:**

My current recommendation is reject.
My reasons are the following:

1. The paper has some good theoretical results and empirical results. However, the current paper seems more like a rough composition of these results and does not have a very clear story for me to follow. I did not see a very clear motivation of the problem.

2. I do not think problem 1 is a novel problem formulation. Therefore the paper should discuss its relation with some closely related problems like meta-learning.

3. Technical novelty is not clear to me.

Overall, I think the current draft requires a big amount of modifications to make it reaching the acceptance level.

---

> ### Author Response · Authors · 2022-11-14
> **To Reviewer KXgD(2/2)**
>
> > The paper considers the case where all the $f_i$’s are strongly convex, which is the most simplified case. That is fine. But the paper did not explicitly discuss the unique technical challenges under this setting. It is hard to justify the technical novelty given the current writing. I do believe the augmented Lagrangian term is novel but it is hard to judge given the current writing. Besides, the algorithmic design section is also hard for me to spot what is the novelty. I think the authors should emphasize the technical challenges and novelty.
>
> First, it is not straightforward to give the augmented Lagrange function. Augmented Lagrange functions are well-studied when equality constraints are linear. But in our problem, the equality constraints are non-linear. Further, adding square error as what the augmented Lagrange method does can not make the Lagrange function strongly convex, and then the min-max optimization process may not converge to the KKT point. So, instead of adding a square term, we add the original function with which the augmented Lagrange function becomes strongly convex. Then, we can directly apply the min-max optimization process and write down the algorithm.
>
> However, it becomes an issue for analyzing the convergence of the proposed algorithm. Usually, people face stochastic only for the primal optimization process, but in our scenario, both primal and dual steps face stochastic gradient issues. Further, different from linear constraints, the Lipschitz is fixed for all $\lambda$. We have to bound $\lambda$ to control the Lipschitz constant. Since, with the stochastic update of $\lambda$, it is hard to give a uniform bound of $\lambda$ almost surely, we add a constraint on $\lambda$, which gives the same KKT point as the problem (1) and controls Lipschitz constant for the optimization process.
>
> Last, when the update of $\lambda$ faces constraints, and randomness, it is more complicated than it is in the traditional augmented Lagrange method.

---

> ### Author Response · Authors · 2022-11-14
> **To Reviewer KXgD(1/2)**
>
> Thanks for your valuable comments. We respond to your questions as follows:
> > Not well-motivated: I indeed believe problem eq (1) is a very meaningful problem. But I do not think there is enough motivation in this paper.
> > + i) the original motivation is that one would like to train a model parameter that is good for some global task $f_0$. This makes sense but why not directly optimize for $f_0$? The reason is (my understanding) there is not enough data for optimization w.r.t. $f_0$ directly and therefore we need some pre-training using all local clients’ data. Motivation like this should be emphasized.
> >+  ii) under the same motivation, equation 1 is only one of the many ways to formulate the problem. It would be necessary to discuss alternative problem formulation as well. For example, why not first pretrain on local clients data using federated learning and then fine-tuning on few-shot data from task $f_0$?
> >+ iii) When I first read, it was not immediately unclear to me why the estimation of Hessian in the previous methods is hard to implement under this paper’s setting due to the large amount of data. My understanding is one cannot get an unbiased estimator of the Hessian. Issues like this should also be discussed in more depth because it motivates the paper.
> >+ Motivate problem formulation given by equation (1). Especially, what is the relation between this problem and meta-learning? In Meta-learning, one would like to find a model parameter that is good for (possibly unseen) downstream tasks. This is clearly very related with this problem considered in this paper.
>
> i) Our motivation is that when many devices come into federated learning to contribute their data for the global model, it is unclear whether their data is useful for real applications. For example, 7 nodes with absolute noise in the experiment refer to the classification task. In federated learning, people usually use the same weight for each device or assign weight according to the number of data points.
>  Mohri et al. (2019) proposed agnostic federated learning to deal with this kind of naive weights, but they focused on the most difficult data set among all devices. Instead of guessing what task the global model needs to solve in our setting, we introduce a validation set to characterize the task. Similar to hyperparameter tuning, where we can view $x$ as hyper-parameters and $w$ as the model parameters, it directly gives the bilevel formulation.
>
> ii) We are sorry that we do not understand the algorithm that pretrains and fune-tunes. As in the setting, some devices give data with large noise (e.g., random data in the experiment). Finetuning with those data will decrease performance in general.
>
> iii) Every bi-level method needs to estimate the inverse of Hessian. It is because with the assumption of strong convexity of the lower level optimization problem,  the gradient of $x^{(i)}$  is $\nabla f_0(w^*(x))^\top(\sum_{i=1}^N x^{(i)} \nabla^2 f_i(w^*(x)))^{-1}\nabla f_i(w^*(x))$. When it comes to a large-scale problem, it is not easy to obtain either $\nabla f_i(w)$ or $\nabla^2 f_i(w)$ exactly, let alone obtaining the inversion of Hessian. In the literature, there are two major methods for estimating the inverse of Hessian. One of the method (represented by BSA), people use $\sum_{k=0}^K \alpha (I-\alpha H)^kv$ to  approximate $H^{-1}v$, where $H$ can be viewed as Hessian matrix, and $v$ can be viewed as some vector. The other method is based on gradient descent update ($w_{t+1} = w_t - \alpha \sum_{i=1}^Nx^{(i)}\nabla f_i(w_t)$) and tries to use $\frac{\partial f_0(w_K)}{\partial x^{(i)}} = \nabla f_0(w_t)^\top (\alpha\sum_{k=1}^K (\Pi_{t=k}^K(I-\alpha \sum_{i=1}^N x^{(i)} \nabla^2 f_i(w_k)) \nabla f_i(w_k))$, while the calculation can be done by auto-differential in the deep learning framework. Different from these two methods, we approximate the inverse of Hessian with the dual variable $\lambda$. The benefit of using a dual variable is that with the smoothness assumption of Hessian, $\lambda$ in the previous round can be a better initialization for approximation, while the other two methods have to restart approximation according to their formula. When variables in upper optimization is unconstrained, with accurate approximation such that the direction is a descent direction, the algorithm can converge to the optimal solution. However, when upper optimization has some constraints, even if it is convex, the descent direction can not guarantee convergence to the optimal point for convex optimization. Thus, when $x$ has constraints, we need a more accurate approximation. As $\lambda$ can have some good initialization during the optimization process, we believe the primal-dual method can achieve better performance than the bi-level optimization methods in the previous work.

---

### Official Review · Reviewer_Yrj6 · 2022-10-20

**Confidence:** 4
**Correctness:** 1
**Technical Novelty And Significance:** 3
**Empirical Novelty And Significance:** 3
**Recommendation:** 3

**Clarity, Quality, Novelty And Reproducibility:**

The clarity should be improved, the idea of the paper is novel to my knowledge, reproducibility is not clear.

**Strength And Weaknesses:**

The major issue of this paper is the writing and presentation, I list some writing issues I found when reading the paper:
- The first sentence is exactly the same as the first sentence of the abstract.
- The citation format is not very careful, should use \citep in some places, for example in Remark 1.
- What is $w$ and $x$ in equation 1? Although these information can be recovered in the following context, it is still better to explain equation more clearly after equation 1.
- Second paragraph of introduction: propose and algorithm -> propose an algorithm
- Second paragraph of introduction: gradient calculate -> gradient calculation
- Second paragraph of introduction: will involved -> will involve
- Section 2.2, One Franceschi et al.., the sentence is informal
- Section 2.2, Different from above work -> works
- The paragraph below Proposition 1, For given x -> For given $x$.
- What is "LICQ" in Proposition 2.
- The paragraph below Proposition 3, constraint set ad $\Lambda$ -> constraint set $\Lambda$.
- (A1), should be $f_0, \ldots, f_N$ have $L_1$ gradient and are lower bounded by $\underline{f}$.
- (A4) looks weird, it looks like a conclustion instead of an assumption.
- (A5) Better to express it with a math formula.
- Proposition 4, when ..., then the stationary point....
- The paragraph below Theorem 1, theorem1 -> theorem 1, Then, We -> Then, we
- Remark 2, bilevel works, too informal, should be previous works on bilevel optimization.
....

Some other issues:
- It is not clear to me why the bilevel formulation (problem 1) is important. Seem like an intuitive problem formulation without much theory behind it?
- The author(s) stated that $f_0$ depends on the validation set, which is weird. In ML, validation set usually used for hyper-parameter tunning and should not be involved in the training.

**Summary Of The Paper:**

The author(s) formulated the objective of the federated learning problem as a bi-level optimization problem. The author(s) proposed an algorithm that solve the bi-level optimization problems with stochastic gradient oracle. Convergence analysis of the proposed algorithm is given and experiments on toy and real-world datasets are conducted to evaluate the proposed method.

**Summary Of The Review:**

The writing and presentation of the paper should be improved. It is not clear if the bi-level formulation is important, evidences (either theoretical or empirical) for the problem formualtion is not given. The current version of the manuscript seems not good enough to appear in ICLR.

---

> ### Author Response · Authors · 2022-11-14
> **To Reviewer Yrj6**
>
> Thanks for your valuable comments. We respond to your questions as follows:
> >The major issue of this paper is the writing and presentation, I list some writing issues I found when reading the paper.
>
> Sorry for those typos. We have already corrected them.
>
> > It is not clear to me why the bilevel formulation (problem 1) is important. Seem like an intuitive problem formulation without much theory behind it?
> > The author(s) stated that $f_0$ depends on the validation set, which is weird. In ML, validation set usually used for hyper-parameter tunning and should not be involved in the training.
>
> Similar to the hyper-parameter tuning, we can view $x$ as the hyper-parameters in federated learning, which give different coefficients to different devices. We can view $w$ as the model parameters. Then similar to hyper-parameter tunning, we can directly formulate the bi-level optimization problem. Based on the formula of bi-level optimization, it is easy to see that the optimal $x$ is the optimal coefficient that combines different nodes.
>
> Further, our contribution is in the following. First, we introduce bi-level optimization into federated learning to distinguish devices with low data quality (e.g., for 7 noisy devices in the experiment). We propose a new algorithm to solve bi-level optimization that performs better than previous work. Last, for non-linear constraints optimization, we give a new augmented term that can obtain the same KKT point and make the algorithm converge.
>
> To make the idea clear, we rewrite the Introduction section.

---

### Official Review · Reviewer_BDrt · 2022-10-27

**Confidence:** 3
**Correctness:** 4
**Technical Novelty And Significance:** 3
**Empirical Novelty And Significance:** 2
**Recommendation:** 6

**Clarity, Quality, Novelty And Reproducibility:**

The clarity of the paper can be improved. Specifically it was difficult for me to understand 5 lines before problem (1) when I read that part first. I recommend that authors express that part in a simpler and easier to understand way. Also, there are some typos in the paper. For example in line 2 of section 2 on page 2, there is a typo. The proposed algorithm is novel. However, I could not check the proofs in the paper carefully.

**Strength And Weaknesses:**

Strength:

1. The paper proposes a new bi-level optimization algorithm for federated learning to find the optimal parameter with respect to local objectives.
2. The paper provides the convergence guarantees for the proposed algorithm.

Weaknesses:

1. Using the proposed algorithm, the server has to perform more computations than other well-known existing algorithms. One of the benefits of federated learning is to push the computations from central servers to edge devices with good computational capability. This can adversely affect the applicability of the proposed algorithm compared to fedavg-based existing ones especially when the number of clients is large.
2. Experimental results presented in the paper is not enough to show the effectiveness of the proposed algorithm. The paper mainly compares the proposed algorithm with bi-level optimization methods. However, I believe that it is better to compare the proposed algorithm with other fedavg-based algorithms in terms of both training time and test data accuracy. Also agnostic federated learning and personalized federated learning are closely connected to the study of the paper and it would be interesting if the paper compares the proposed algorithm with some personalized algorithms as well as agnostic federated learning.

**Summary Of The Paper:**

This paper considers training a global model on the server using updates receive from clients. The paper proposes that clients and the server collaborate to solve a bi-level optimization problem. This enables the server to learn the global model using the optimal parameter and combination of the local objectives.

**Summary Of The Review:**

In summary, the paper proposes a novel federated learning algorithm with theoretical guarantees. However, the empirical study in the paper is not enough to show its effectiveness.

---

> ### Author Response · Authors · 2022-11-14
> **To Reviewer BDrt**
>
> Thanks for your valuable comments. We respond to your questions as follows:
> > Using the proposed algorithm, the server has to perform more computations than other well-known existing algorithms. One of the benefits of federated learning is to push the computations from central servers to edge devices with good computational capability. This can adversely affect the applicability of the proposed algorithm compared to fedavg-based existing ones especially when the number of clients is large.
>
> Compared to FedAvg, for the inner loop, in the server, we compute one more gradient of the model ($\nabla f_0(w)$), which will not increase computation when the number of clients becomes large. The rest calculation is just adding up the vectors sent from the devices. The highly cost computation may come from the update of $x$. For simplicity, we calcuate the update of $x$ in the server. But we can calculate the update of $x^{(i)}$ in i-th device and send the update of $x^{(i)}$ to the server, and this only needs to transmit one real value from the devices and will not increase communication cost much compared to the update of $w$. We add a remark after the algorithm to explain the distributed calculation of the update of $x$.
>
> >Experimental results presented in the paper is not enough to show the effectiveness of the proposed algorithm. The paper mainly compares the proposed algorithm with bi-level optimization methods. However, I believe that it is better to compare the proposed algorithm with other fedavg-based algorithms in terms of both training time and test data accuracy. Also agnostic federated learning and personalized federated learning are closely connected to the study of the paper and it would be interesting if the paper compares the proposed algorithm with some personalized algorithms as well as agnostic federated learning.
>
> We add experiments that are compared to Agnostic Federated learning and change the figure in the experiment section. For personalized federated learning, it is not fair to compare them with our formula because algorithms for personalization are to fit each local device with their local models, while we aim to fit the global data. Even if we can make the global data as one of the local data, we can build a personalized model for the specific device. It is unfair because we introduce a validation set while the other methods do not.
>
>
> >The clarity of the paper can be improved. Specifically it was difficult for me to understand 5 lines before problem (1) when I read that part first. I recommend that authors express that part in a simpler and easier to understand way. Also, there are some typos in the paper. For example in line 2 of section 2 on page 2, there is a typo. The proposed algorithm is novel. However, I could not check the proofs in the paper carefully.
>
> Sorry for the confusion. We change the introduction section of the paper.

---

### Official Review · Reviewer_Ycsn · 2022-10-31

**Confidence:** 4
**Correctness:** 3
**Technical Novelty And Significance:** 3
**Empirical Novelty And Significance:** 2
**Recommendation:** 5

**Clarity, Quality, Novelty And Reproducibility:**

I think the writing of the paper is good as it is easy to follow. The theoretical results in the paper do have novelty compared to existing works. However, I need to see more discussion on how the new results stand compared with existing one, and whether the new approach is applicable in federated learning setting. The paper is associated with code for reproducibility which is a plus.

**Strength And Weaknesses:**

Strengths:
- The primal-dual approach presented in the paper appears to be new compared to existing work.
- Theretical analyses are provided. I have gone through the key steps in the proof and find no problem but there's a chance I miss something.
- Numerical experiments on synthetic and vision datasets show that the new appoach obtain promising performance.

Weaknesses:
- I have one major concerns which is the applicability of the proposed method in federated learning. I am not sure if the authors design the new primal-dual algorithm as a federated learning algorithm or not. If it is, then the local devices should send back their local models instead of the stochastic gradient/hessian. Therefore, I believe their algorithm instead is somewhat a distributed learning algorithm, not specifically for federated learning. I notice that only in this paper the bi-level optimization is applied to federated learning while others do not mention it.
- Another concern is about the metric of evaluation. In federated learning, the main challenge is the communication efficiency (or the number of bits communicated between clients and server). As a results, other federated learning methods use performance over communication round or communicated bits as the metric of comparison. I think the metric in section 5 should be adjusted to illustrate this. Also, the payload of the primal-dual apporach can be twice as much as others such as BSA in 1 iteration so it might not be fair to use iteration as in the current paper.

Minor comment:
- Typo on page 9: when all local devices participant in -> participate in

Suggestion for improvement:
- I think it would be good to discuss the convergence rate obtained by the primal-dual approach vs existing work as I need to go back to their original paper to retrieve their rates.
- I hope to see clarification whether the new algorithm is for federated learning since it does not follow federated learning framework where clients only send back local model instead of local gradients.
- The metric of comparison for numerical experiments may need to be revised to match standard comparison in federated learning.

**Summary Of The Paper:**

The paper focuses on solving the local coefficient learning in federated learning as a bi-level optimization problem. Instead of directly solving the bi-level problem, they reformulate it as a convex-concave problem where they propose a double loop algorithm to solve it. The authors show that the proposed algorithm achieves the sample complexity of $\mathcal{O}(\varepsilon^{-4})$ to arrive at a $\varepsilon$-stationary point. Experiments on synthetic and vision datasets are conducted to illustrate the performance of the new primal-dual appoach versus existing bi-level algorithms.

**Summary Of The Review:**

As mentioned above, I believe the paper does have contribution in algorithmic design to solve the bi-level optimization but it is not clear whether the new algorithm is suitable for federated learning. Also, the comparison metric using performance in iteration may not be fair as the communication payload is different among the algorithms used in Section 5.

---

> ### Author Response · Authors · 2022-11-14
> **To Reviewer Ycsn**
>
> Thanks for your valuable comments. We respond to your questions as follows:
> > I have one major concerns which is the applicability of the proposed method in federated learning. I am not sure if the authors design the new primal-dual algorithm as a federated learning algorithm or not. If it is, then the local devices should send back their local models instead of the stochastic gradient/hessian. Therefore, I believe their algorithm instead is somewhat a distributed learning algorithm, not specifically for federated learning. I notice that only in this paper the bi-level optimization is applied to federated learning while others do not mention it.
>
> + There are still many papers in federated learning that transmit local updates instead of the local model (e.g.
>  Karimireddy et al. (2020)). Transmitting stochastic gradient/hessian is one way to send local updates.
> + We formulate the local coefficient problem as a bi-level optimization and propose a primal-dual algorithm to solve it in the federated setting. Both of these are novel in the federated learning community.
>
> > Another concern is about the metric of evaluation. In federated learning, the main challenge is communication efficiency (or the number of bits communicated between clients and the server). As a result, other federated learning methods use performance over communication rounds or communicated bits as the metric of comparison. I think the metric in section 5 should be adjusted to illustrate this. Also, the payload of the primal-dual apporach can be twice as much as others such as BSA in 1 iteration so it might not be fair to use iteration as in the current paper.
>
> We compare fairly with other bi-level optimization methods. For BSA, it also needs to communicate two vectors. One is to update variable $w$, and the other is to update variable $x$. Our algorithm uses two vectors to update variable $w$ and variable $\lambda$. Then, we update the variable $x$ on the server side with updated variables $w$ and $\lambda$. Thus, we have the same communication cost as the other bi-level methods in one iteration. We have added a discussion on communication costs in the experimental section.
>
> > I think it would be good to discuss the convergence rate obtained by the primal-dual approach vs existing work as I need to go back to their original paper to retrieve their rates.
>
> For other bi-level methods, they derive the convergence without constraint on $x$, where we have non-negative and norm constraints. So our convergence rate can not be compared with others.

---

### Decision · Program_Chairs · 2023-01-20

**Decision:**

Reject

**Justification For Why Not Higher Score:**

Empirical evaluation and theory currently do not support all of the claims made in the paper well enough to justify acceptance.
Several aspects of the problem formulation could be better motivated and justified to convince readers of the relevance of this formulation.

**Justification For Why Not Lower Score:**

N/A

**Metareview: Summary, Strengths And Weaknesses:**

This paper proposes an alternative formulation to federated learning which introduces an additional weighting variable over clients that is optimized at the server via a bi-level optimization.

The bi-level formulation is interesting and novel, and the theory was one strength of this paper.

However there were also several noted weaknesses:
* There should be further comparison with other federated learning methods and especially other personalized federated learning methods that have a closely related formulation.
* Some aspects of the proposed formulation and approach seem strong and need better justification/motivation, such as assuming the existence of a validation set at the server that is reflective of the data distribution at all clients. (If one has such data, why train using FL as opposed to just training directly on that data at the server? Why not use that data in other ways?)
* Several suggestions were also made to improve the presentation and clarity of the paper.

**Summary Of Ac-Reviewer Meeting:**

n/a